



# Stalagmite carbon isotopes suggest deglacial increase in soil respiration in Western Europe driven by temperature change

Franziska A. Lechleitner[1,2*], Christopher C. Day[1], Oliver Kost[3], Micah Wilhelm[4], Negar Haghipour[3,5], Gideon M. Henderson[1], and Heather M. Stoll[3]

[1]Department of Earth Sciences, University of Oxford, South Parks Road, OX1 3AN Oxford, UK

[2]Department of Chemistry and Biochemistry & Oeschger Centre for Climate Change Research, University of Bern, Freiestrasse 3, 3012 Bern, Switzerland

[3]Department of Earth Sciences, ETH Zurich, Sonneggstrasse 5, 8006 Zürich, Switzerland

[4]Swiss Federal Institute for Forest, Snow and Landscape Research, Zürcherstrasse 111, 8903 Birmensdorf, Switzerland

[5]Laboratory for Ion Beam Physics, ETH Zürich, Otto-Stern-Weg 5, 8093 Zürich, Switzerland

*Correspondence to*: Franziska A. Lechleitner (franziska.lechleitner@dcb.unibe.ch)

**Abstract.** The temperate region of Western Europe underwent dramatic climatic and environmental change during the last deglaciation. Much of what is known about the terrestrial ecosystem response to deglacial warming stems from pollen preserved in sediment sequences, providing information on vegetation composition. Other ecosystem processes, such as soil respiration, remain poorly constrained over past climatic transitions, but are critical for understanding the global carbon cycle and its response to ongoing anthropogenic warming. Here we show that speleothem carbon isotope ($\delta^{13}C_{spel}$) records may retain information on local soil respiration, and allow its reconstruction over time. While this notion has been proposed in the past, our study is the first to rigorously test it, using a combination of multi-proxy geochemical analysis ($\delta^{13}C$, Ca isotopes, and radiocarbon) on three speleothems from Northern Spain, and quantitative forward modelling of processes in soil, karst, and cave. Our study is the first to quantify and remove the effects of prior calcite precipitation (PCP, using Ca isotopes) and bedrock dissolution (using the radiocarbon reservoir effect) from the $\delta^{13}C_{spel}$ signal to derive changes in respired $\delta^{13}C$. Coupling of soil gas $pCO_2$ and $\delta^{13}C$ via a mixing line describing diffusive gas transport between an atmospheric and a respired end member allows modelling of changes in soil respiration in response to temperature. Using this coupling and a range of other parameters describing carbonate dissolution and cave atmospheric conditions, we generate large simulation ensembles from which the results most closely matching the measured speleothem data are selected. Our results robustly show that an increase in soil $pCO_2$ (and thus respiration) is needed to explain the observed deglacial trend in $\delta^{13}C_{spel}$. However, the $Q_{10}$ (temperature sensitivity) derived from the model results is higher than current measurements, suggesting that part of the signal may be





related to a change in the composition of the soil respired $\delta^{13}C$, likely from changing substrate through increasing contribution from vegetation biomass with the onset of the Holocene.

## 1 Introduction

The last deglaciation was a period of profound global climate change. Between 22 and 10 ka BP (ka: thousands of years, BP:
"before present", with the present referring to 1950 CE), global mean surface air temperatures increased by up to ~6°C (Tierney et al., 2020), leading to the disintegration of the large Northern Hemisphere ice sheets and a consequent rise in global sea level by ~80-120 m (Lambeck et al., 2014). On land, shifts in ecosystem types and productivity accompanied the deglacial climate change, with repercussions on the terrestrial carbon cycle and the release of greenhouse gases to the atmosphere (Clark et al., 2012). The temperate region of Western Europe was particularly affected by large and latitudinally diverse environmental
changes during the last deglaciation, driven by its proximity to the Scandinavian Ice Sheets and the North Atlantic (Moreno et al., 2014). Over the entire region, terrestrial paleoclimate records indicate a transition from colder to warmer climatic conditions, punctuated by millennial-scale events which closely match the Greenland ice core record (Genty et al., 2006; Moreno et al., 2014). Pollen records from Western Europe reveal a general deglacial trend from grassland steppe and tundra ecosystems towards landscapes dominated by temperate forest, and provide evidence for remarkably rapid ecosystem response
to temperature changes on millennial scales over the last glacial (Fletcher et al., 2010).

Speleothem carbon isotope ($\delta^{13}C_{spel}$) records from the temperate region of Western Europe are often clearly correlated to regional temperature reconstructions during the last glacial (Genty et al., 2003) and the deglaciation (Baldini et al., 2015; Denniston et al., 2018; Genty et al., 2006; Moreno et al., 2010; Rossi et al., 2018; Verheyden et al., 2014) (Fig. 1). These records are also highly consistent in timing, amplitude, and absolute $\delta^{13}C_{spel}$ values amongst each other, pointing towards a
regionally coherent mechanism driving the response to the temperature increase. Early on, Genty et al. (2006, 2003) suggested that the temperature sensitivity of $\delta^{13}C_{spel}$ in Western Europe was likely related to the response of vegetation and soil respiration to climate warming. Higher concentrations of respired $CO_2$ in the soil lower its $\delta^{13}C$ signature, due to the increase of strongly fractionated organic carbon in the system. Speleothems can capture this change as they are fed by dripwater, which equilibrates with soil $pCO_2$ before proceeding to the dissolution of carbonate bedrock. This mechanism could lead to the observed
transitions from higher $\delta^{13}C_{spel}$ during colder periods to lower $\delta^{13}C_{spel}$ during warmer periods, and may provide a means to quantify past changes in soil respiration, an elusive parameter in the global carbon cycle (Bond-Lamberty and Thomson, 2010). However, formal testing of this mechanism has so far not been attempted, mainly because of the numerous and complex processes that influence $\delta^{13}C_{spel}$ (Fohlmeister et al., 2020).

Speleothem carbon can originate from three sources: atmospheric $CO_2$, biogenic $CO_2$ from autotrophic (root and rhizosphere)
and heterotrophic (soil microbial) soil respiration (from here onwards jointly referred to as "soil respiration"), and the carbonate bedrock itself. Recent research has additionally suggested that deep underground reservoirs of carbon ("ground air"; Mattey





et al., 2016) or deeply rooted vegetation (Breecker et al., 2012) may play a significant role in the karst carbon cycle. The relative importance of these different sources on $\delta^{13}C_{spel}$ is modulated by hydroclimate and temperature. This can occur as a propagation of a biosphere response to climate change, e.g., changes in vegetation composition (Braun et al., 2019), changes

in soil respiration (Genty et al., 2003), and changes in soil turnover rates (Rudzka et al., 2011). Secondly, $\delta^{13}C_{spel}$ can be modulated by changes in karst hydrology, i.e., carbonate bedrock dissolution regime (Hendy, 1971). Thirdly, compounded changes in hydrology and cave atmospheric $pCO_2$ can lead to prior calcite precipitation (PCP) during carbonate precipitation (Fohlmeister et al., 2020). Altitudinal transects in caves in the European Alps have shown that changes in soil respiration, vegetation, and temperature have a tractable effect on speleothem fabrics, stable oxygen isotope ratios, and $\delta^{13}C_{spel}$ (Borsato

et al., 2015). So far, it has not been possible to quantify the relative importance of these processes on $\delta^{13}C_{spel}$ records, but this quantification is a crucial step towards disentangling the effects of soil respiration from other influences, and to evaluate the potential of $\delta^{13}C_{spel}$ as a paleo-soil respiration proxy. Here, we generate a multi-proxy dataset from three stalagmites from northern Spain and use quantitative forward modelling to show that changes in soil respiration can explain much of the observed deglacial trend in Western European $\delta^{13}C_{spel}$. Our approach is the first to leverage differing proxy sensitivities to

quantitatively model key environmental parameters, in particular soil gas $pCO_2$, allowing us to estimate the total temperature sensitivity of soil respiration ($Q_{10}$), including the effect of changing vegetation communities.

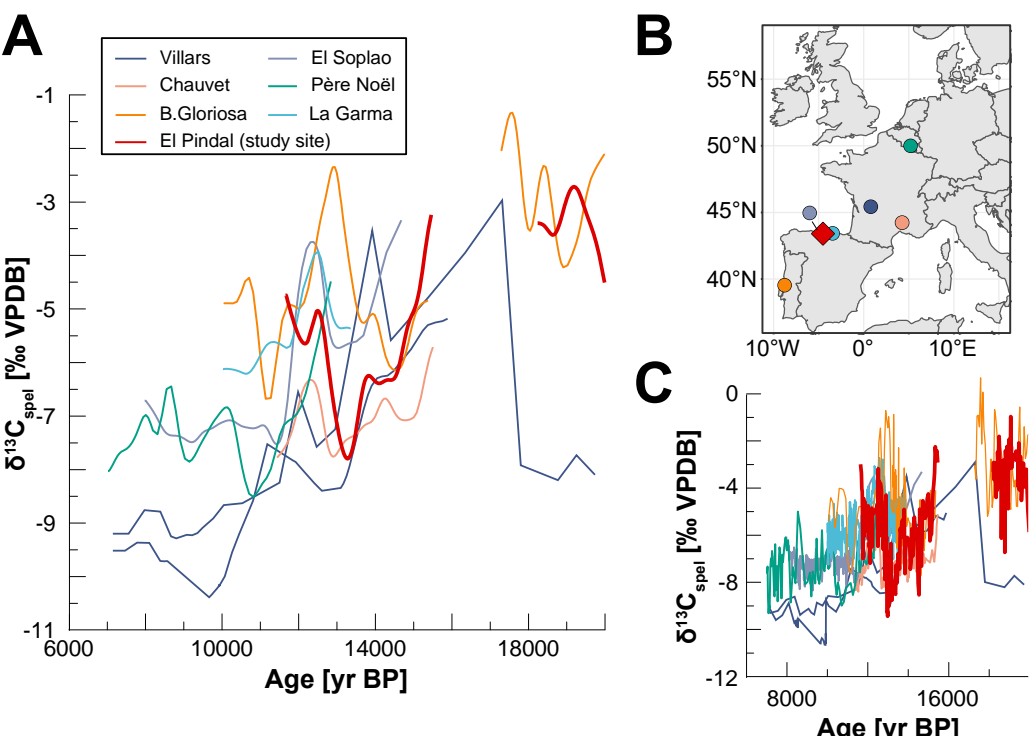

Figure 1: Speleothem $\delta^{13}C$ records covering the last deglaciation in temperate Western Europe. A – Records vs age, colour coded by cave. Villars Cave – stalagmites Vil-stm11 (Genty et al., 2006) and Vil-car-1 (Wainer et al., 2011); Chauvet Cave –



stalagmite Chau-stm6 (Genty et al., 2006); El Pindal Cave – stalagmite Candela (Moreno et al., 2010); La Garma Cave –
       stalagmite GAR-01 (Baldini et al., 2015); El Soplao Cave – stalagmite SIR-1 (Rossi et al., 2018); Père Noël Cave – stalagmite
       PN-95-5 (Verheyden et al., 2014); Buraca Gloriosa – stalagmite BG6LR (Denniston et al., 2018). All stalagmite data was
       extracted from the SISAL database, version 2 (Comas-Bru et al., 2020b, 2020a). Shown here is the millennial-scale trend in
       the records, calculated using a gaussian kernel smoother (nest package in R, Rehfeld and Kurths, 2014). B – Cave locations.

C – Original (not filtered) records.

## 2 Study site and samples

El Pindal and La Vallina caves are located ~30 km apart on the coastal plain in Asturias, NW Spain, at 23 and 70 m a.s.l.
respectively (43°12'N, 4°30'W, Fig. 1). Both caves developed in the non-dolomitic, Carboniferous limestones of the

Barcaliente formation, with an overburden of 10-35 m of bedrock for El Pindal Cave and 10-20 m for the gallery in which
       samples were collected in La Vallina.

Current climate in northern Spain is characterised by temperate maritime conditions, with clear precipitation seasonality, but
no summer drought (Peinado Lorca and Martínez-Parras, 1987). The region is strongly affected by North Atlantic climate
conditions, in contrast to the rest of the Iberian Peninsula, where North Atlantic and Mediterranean influences persist (Moreno

et al., 2010). Both caves are affected by similar climatic conditions, with ~1250 mm/yr annual precipitation (Stoll et al., 2013),
       and maximum precipitation occurring in November (140 mm/month) (AEMET meteorological stations at Santander and
       Oviedo, period 1973-2010; AEMET, 2020). Due to the proximity to the coast, temperature exhibits a clear but modest
       seasonality, with averages of 9°C for winter months (December-February), and 20°C for summer months (June-September)
       (AEMET meteorological station at Santander, period 1987-2000; AEMET, 2020). For the last deglaciation, quantitative

estimates of temperature can be derived from marine records from the western and southern Iberian Margins. These likely give
       a reasonable estimate of the deglacial temperature change in caves on the coastal plain, as the region's modern seasonal cycle
       displays similar amplitude to sea surface temperatures (Stoll et al., 2015). Minimum average temperatures are reconstructed
       for Heinrich event 1 (H1; 18-15 ka BP) and are ~8°C cooler than those of the Holocene Thermal Maximum (~8 ka BP; Darfeuil
       et al., 2016).

Previous monitoring data from the two caves reveals seasonal variations in cave $pCO_2$ driven by external temperature variations
       (Moreno et al., 2010; Stoll et al., 2012). Both caves are well ventilated in the cold season with close to atmospheric $pCO_2$
       values, but feature elevated $CO_2$ concentrations during the warm summer season (Stoll et al., 2012). The caves are covered by
       thin (<1m deep) and rocky soils, and modern vegetation is strongly impacted by Late Holocene land use change, including
       deforestation of native *Quercus ilex* (evergreen oak) for lime kilns above El Pindal Cave, and discontinuous pasture maintained

by cycles of burning above both caves. At present, the vegetation above the two caves includes pasture and gorse shrub (*Ulex*),
       but in some areas above El Pindal Cave, the recent abandonment of pastures has permitted the return of patches of native



*Quercus ilex* forest. Above La Vallina Cave, pastures are interspersed with native oak (*Quercus*) and planted groves of *Eucalyptus*, the roots of which penetrate the cave in points directly beneath the tree groves.

Candela is a calcitic stalagmite that grew ~500 m inside El Pindal Cave and was not active at the time of collection (Moreno

et al., 2010). Previous investigations revealed that the stalagmite grew between ~25 – 7 ka BP and provide high resolution stable isotope and trace element records (Moreno et al., 2010), as well as $^{14}C$ measurements between 15.4 – 8.8 ka BP (Rudzka et al., 2011). Growth of Candela is strongly condensed between 18-15.5 ka BP and 11-9 ka BP (Stoll et al., 2013). Stalagmite Laura is from El Pindal Cave, while Galia grew in La Vallina Cave. Both Laura and Galia are also composed of calcite. Previous U-Th dating on Galia revealed intermittent growth between 60 and 4 ka BP (Stoll et al., 2013), including a short

growth phase at 26 ka BP which together with the Holocene growth is sampled here. Laura grew between 16.1-14.2 ka BP, covering the H1-Bølling-Allerød (BA) interval.

## 3 Methods

### 3.1 Geochemical measurements

To minimise sampling bias, samples from all three stalagmites were drilled from the same locations for all geochemical analyses using either a hand-held drill or a semi-automated high precision drill. An aliquot of the collected powder was used each for U-Th dating, $\delta^{13}C$, $^{14}C$, and $\delta^{44/40}Ca$. In the case of Candela, where a few U-Th dates were available from previous investigations (Moreno et al., 2010), powders for the remaining proxies were drilled from the same sampling holes. Additional paired MC-ICPMS U-Th dates from all three stalagmites are detailed elsewhere (Stoll et al., in review).

For stable carbon isotopes, an aliquot of powder was analysed on a ThermoFinnigan GasBench II carbonate preparation device at the Geological Institute, ETH Zurich, following the procedure by Breitenbach and Bernasconi (2011). Measurement runs were evaluated using an in-house standard (MS2) that has been linked to NBS19 and the external standard deviation ($1\sigma$) for $\delta^{13}C$ is smaller than 0.08 per mil (‰). Isotope values are expressed in ‰ and referenced to the Vienna Pee Dee belemnite standard (VPDB).

Radiocarbon measurements were performed at the Laboratory for Ion Beam Physics, ETH Zurich, using a MICADAS accelerator mass spectrometer (AMS; Synal et al., 2007) coupled to a gas ion source (GIS; Fahrni et al., 2013). Carbonate powders (~1 mg) were dissolved in 85% $H_3PO_4$ and the resulting $CO_2$ gas was directly injected into the GIS. Quality control of the AMS measurements was ensured by measuring Oxalic acid II (NIST SRM 4990C), IAEA C-2 as a carbonate standard, and IAEA C-1 as carbonate blank, and measurement precision was better than 10‰. We use the $^{14}C$ reservoir effect ("dead

carbon fraction", DCF), which quantifies the amount of fossil carbon incorporated in the speleothems and serves as a tracer for changes in karst hydrology or mean soil carbon age (Genty et al., 2001). The DCF is calculated as the normalized difference between the atmospheric $^{14}C$ activity ($F^{14}C$; Reimer, 2013) at the time of speleothem deposition (defined through the independent U-Th chronology), and the speleothem $^{14}C$ activity corrected for decay. Using paired U-Th and $^{14}C$ ages has the





advantage of minimizing uncertainty from age modelling interpolation techniques. To account for the uncertainty in matching

the speleothem chronology with the atmospheric [14]C record (IntCal13 calibration curve; Reimer et al., 2013) the atmospheric record was interpolated to yearly resolution and matched to 10,000 simulated speleothem ages for each U-Th dating point. The average and standard deviation from these ensembles were then used for the final DCF calculation and uncertainty propagation. Samples for Ca-isotope analysis were taken from the stalagmites and from three pieces of bedrock overlying both caves. Combined bedrock and stalagmite Ca-isotope analyses allow reconstruction of the Ca-isotopic composition of the initial

growth solution and therefore of the fraction of Ca remaining in solution ($f_{Ca}$) at the point of stalagmite-growth, a quantitative measure for PCP (Owen et al., 2016). Aliquots of $CaCO_3$ (200-650 μg) were dissolved in distilled 2M $HNO_3$. The Ca was purified using an automated Ca-Sr separation method (PrepFAST MC, Elemental Scientific, Omaha, NE, USA). This process separates Ca from Sr, Mg and other matrix elements, to avoid isobaric interferences during multi-collector inductively coupled mass spectrometry (MC-ICP-MS). Ca-isotope ratios were analysed at the University of Oxford using a Nu Instruments MC-

ICP-MS, following the method of Reynard et al. (2011). All solutions were at $10 \pm 1$ ppm concentration, and the samples were measured with standard-sample bracketing. Each sample was analysed a minimum of 5 times. $\delta^{44/40}Ca$ was calculated using $\delta^{44/40}Ca = \delta^{44/42}Ca * ((43.956\text{-}39.963)/(43.956\text{-}41.959))$ (Hippler et al., 2003), and is reported normalised to NIST SRM 915a. Secondary standards HPSnew (in-house standard) and NIST-SRM-915b (purified alongside the samples) were used to determine accuracy and external precision. Measured values for our purified SRM 915b were $\delta^{44/40}Ca = 0.71 \pm 0.06$‰ (2se, n

= 12), which match values obtained by TIMS, $\delta^{44/40}Ca = 0.72 \pm 0.04$‰ (2se; Heuser and Eisenhauer, 2008). Uncertainty on Ca isotope data is quoted as the t-distribution-derived 95% confidence interval on the mean of repeat measurements calculated using either the standard deviation on all repeat measurements on each sample or the standard deviation on all secondary standard analyses, whichever is greater.

## 3.2 Process modelling and sensitivity analysis

Forward modelling of processes occurring in the soil, karst, and cave allowed us to investigate the combination of parameters which would simultaneously simulate $\delta^{13}C_{spel}$, $\delta^{44/40}Ca$, and DCF for each time period sampled. Using $\delta^{44/40}Ca$ and DCF to quantify changes in PCP and bedrock dissolution conditions (open vs closed system), respectively, we can remove these effects from $\delta^{13}C_{spel}$ and derive soil respired carbon and its response to temperature change. We employ the PHREEQC-based,

numerical model CaveCalc (Owen et al., 2018), a tool that enables us to evaluate and combine the effects of PCP and bedrock dissolution quantitatively and systematically. We generate large ensembles of simulations from which we then choose the solutions best fitting the measured proxy data. CaveCalc simulates the equilibration between meteoric water and soil $CO_2$ gas, the subsequent dissolution of the host carbonate rock by this solution, and the degassing of $CO_2$ from the solution in the cave environment that leads to the formation of speleothem carbonate. Key model inputs (Table 1) are the concentration and isotopic

composition of soil $CO_2$ and the degree to which isotopic exchange during carbonate dissolution occurs under open/closed or intermediate conditions (gas volume relative to solution volume), which set the initial saturation state and isotopic composition



of the dripwater. Together with the soil $pCO_2$, the $pCO_2$ of the cave environment is modelled to set the degree of oversaturation the solution will have in the cave, and determines the amount of carbonate which can precipitate before the solution reaches equilibrium. Constraints on the model parameters are given by $\delta^{13}C_{spel}$, $\delta^{44/40}Ca$, and DCF.


Our primary interest is evaluating constraints on soil respiration, soil $pCO_2$ and its isotopic composition. Soil $CO_2$ is a mixture of carbon from respired, atmospheric, and bedrock sources, with its concentration depending mainly on temperature, water content, porosity, and soil depth (Amundson et al., 1998; Cerling et al., 1991). Global regressions find growing season soil $pCO_2$ strongly positively correlated with temperature and actual evapotranspiration (Borsato et al., 2015; Brook et al., 1983).

As soil $pCO_2$ is typically much higher than atmospheric $pCO_2$, $CO_2$ diffuses from the soil along concentration gradients, and its concentration and $\delta^{13}C$ value can be approximated using a mixing line between an atmospheric and a soil carbon end member (which includes carbon from respiration and bedrock dissolution) using the Keeling plot approach (Amundson et al., 1998; Cerling et al., 1991; Pataki et al., 2003). Here we use this relationship to test whether changes in soil respiration can realistically explain the observed deglacial $\delta^{13}C_{spel}$ trend. We define a mixing line with an atmospheric end member given by

pre-industrial atmospheric $pCO_2$ (280 ppmv) and $\delta^{13}C$ (-6.5 ‰). The likely range of values for the soil carbon end member was constrained through monitoring of cave $pCO_2$ and $\delta^{13}C_{cave-air}$ at La Vallina Cave, supplemented by measurements of local atmospheric $pCO_2$ and $\delta^{13}C$ over one year. Monthly $CO_2$ measurements reveal a strong correlation between cave air $pCO_2$ and $\delta^{13}C_{cave-air}$, in particular during the summer, when soil respiration is highest (Fig. 2). We estimated the likely soil carbon end member by linear regression of the summer cave monitoring data, forcing the regression through the atmospheric end member

composition. Additional measurements from the forest around La Vallina Cave tend to show an offset from the regression (Fig. 2), likely due to turbulence and advection effects (measurements were collected during the day when atmospheric disturbances are highest; Pataki et al., 2003). Cave monitoring data from winter months (December-March) were excluded from the regression analysis, as they are likely most affected by $CO_2$ from karst dissolution and dynamic ventilation (Stoll et al., 2012), and less closely reflecting the respired end member. The regression points toward a soil carbon end member with

$CO_2$ concentration of ~7800 ppmv and a $\delta^{13}C$ of -22.9‰ (Fig. 2, Suppl. Table 1). Current vegetation density and soil $pCO_2$ may underestimate Holocene conditions that preceded significant land use alteration, but they provide the best available constraints on the end member. Nonetheless this $pCO_2$ is consistent with predictions based on modern climatology and global regressions of $pCO_2$ from climatic factors (e.g., Borsato et al., 2015; Brook et al., 1983). While cave conservation efforts did not permit extensive monitoring of El Pindal Cave, the proximity and similar conditions to La Vallina Cave allow us to use

this end member for both sites. We explore the effects of a changing isotopic composition of the soil carbon end member, for example through a change in substrate (Boström et al., 2007), by calculating two alternate mixing lines with a respired $\delta^{13}C$ of +/- 3‰ compared to the mixing line defined from the monitoring data (Suppl. Table 1).

The sensitivities of the measured speleothem proxies (DCF, $\delta^{44/40}Ca$, and $\delta^{13}C$) to different processes in the soil-karst-cave system allow us to use them to assess the most realistic coupling between measured $\delta^{13}C_{spel}$ and soil $pCO_2$. For each

combination of soil $pCO_2$ and $\delta^{13}C$ calculated from the mixing lines, changes in mean soil [14]C concentration, dissolution





conditions (termed "gas volume" and indicating the amount of gas that 1L of groundwater solution interacts with; Owen et al., 2018), and cave $pCO_2$ were allowed to vary within realistic bounds (Table 1). These boundary conditions were set based on the available monitoring data, e.g., cave air $pCO_2$ was left to vary between atmospheric and the maximum soil $pCO_2$, modelling the effect of cave ventilation dynamics on the proxies. To test whether the system can also be described without invoking

changes in soil gas $\delta^{13}C$, we performed a second set of experiments ("sensitivity analysis") where all parameters (soil $pCO_2$, soil $^{14}C$, gas volume, cave $pCO_2$) were allowed to vary as before, but soil $\delta^{13}C$ was kept constant at -18‰ (Table 1). For both sets of experiments, each simulation was repeated twice for the Early Holocene (EH, post 10 ka BP) and the Late Glacial (LG, pre 10 ka BP and including deglacial) conditions using published estimates for temperature and atmospheric $pCO_2$ (Darfeuil et al., 2016; Lourantou et al., 2010; Stoll et al., 2012; Table 1).

The model solutions were compared to the measured data from Candela (the stalagmite with the most complete deglacial record), and all solutions matching the measured DCF, $\delta^{44/40}Ca$, and $\delta^{13}C_{spel}$ within a defined interval were extracted. For DCF, the confidence interval of the proxy was chosen, while for $\delta^{13}C_{spel}$ and $\delta^{44/40}Ca$, where measurement uncertainties are much smaller, we defined the threshold at +/- 1.5‰ VPDB and +/- 0.2‰, respectively. Solutions were filtered sequentially for all three proxies, and each possible permutation of the sequences (e.g., DCF -> $\delta^{44/40}Ca$ -> $\delta^{13}C_{spel}$) was calculated. The median

and 25/75 percent quantiles of all filtered solution ensembles are used as final model result. To avoid too many solutions without matches to the data, we selected the 5% simulations closest to the measured proxy value for the sensitivity analysis.

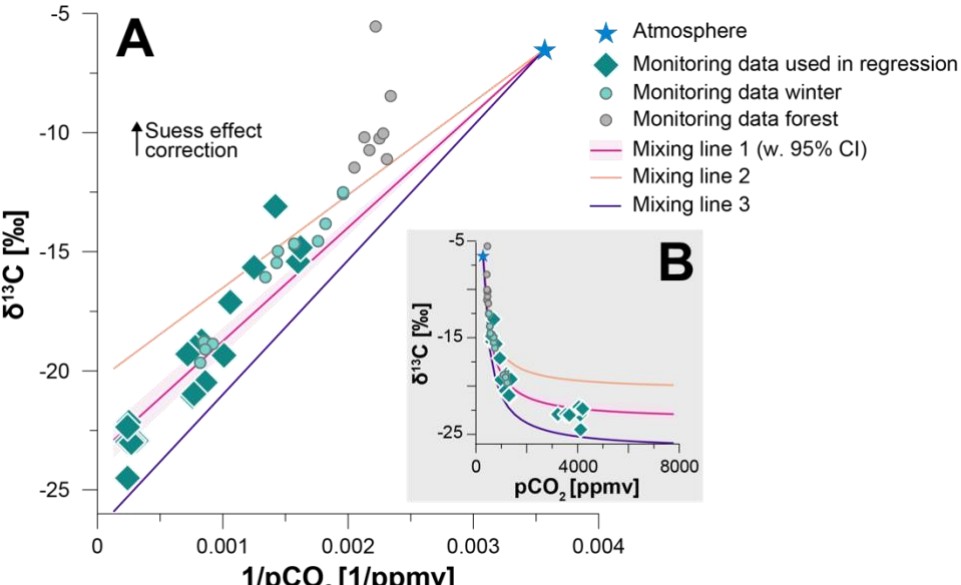

Figure 2: A - Keeling plot of cave monitoring data with respired end member derived from linear regression forced through the atmospheric end member. All data were corrected for the Suess effect to estimate preindustrial values (+1.5 ‰). Mixing

line 1 (red) corresponds to the linear regression between monitoring data and atmospheric end member, but omitting measurements from the forest and cave measurements taken during winter, when the influence from cave ventilation is





strongest (Stoll et al., 2012) and masking the soil carbon end member. Pink shading shows the 95% confidence interval of the linear regression based on the monitoring data. Mixing lines 2 (yellow) and 3 (purple) reflect a change in the respired end member $\delta^{13}C$ by +/- 3 ‰. B – Same as A, but data is shown with respect to $pCO_2$ to emphasize the non-linear relationship

between $\delta^{13}C$ and $pCO_2$.

| Model simulation | Mixing lines | Sensitivity analysis |
|---|---|---|
| T EH (°C) | 12 | 12 |
| T LG (°C) | 4 | 4 |
| atm. $pCO_2$ EH (ppmv) | 260 | 260 |
| atm. $pCO_2$ LG (ppmv) | 180 | 180 |
| soil gas $pCO_2$ (ppmv) | as in mixing line | 280-7800 |
| soil gas $\delta^{13}C$ (‰) | as in mixing line | -18 |
| soil $F^{14}C$ | 100-90 | 100-90 |
| gas volume (L) | 0-500 | 0-500 |
| cave air $pCO_2$ (ppmv) | atmospheric-8000 | atmospheric-8000 |
| host rock Mg (mmol/mol) | 0.6 | 0.6 |
| host rock $\delta^{13}C$ (‰) | 0 | 0 |
| host rock $\delta^{44/40}Ca$ (‰) | 0.58 | 0.58 |

Table 1: Model initial parameters used for the mixing line simulations and sensitivity analysis. Model runs were repeated twice to account for changes in temperature and atmospheric $pCO_2$ between Late Glacial (LG, including deglaciation) and Early Holocene (EH).




## 4 Results

### 4.1 Geochemistry

Both Candela and Galia record a substantial decrease in $\delta^{13}C_{spel}$ between the LG and the EH (Fig. 3). For Candela, $\delta^{13}C_{spel}$ is highest (-2.48 and -4.43‰ VPDB) at 24.9 – 15.4 ka BP, then decreases by about 2‰ with the onset of the BA (14.4 – 12.9 ka

BP). After a short-lived increase back to values of ~-3‰ VPDB at 12.3 ka BP (Younger Dryas, YD), $\delta^{13}C_{spel}$ decreases further to -10 – -7.7‰ VPDB in the EH (8.5 – 7.9 ka BP). In Galia, $\delta^{13}C_{spel}$ is -3.88‰ VPDB during the LGM (26.8 ka BP) and between -9.78 and -8.79‰ VPDB in the EH (8.7 – 4.2 ka BP). Laura covers the time period between 14.3 – 16.1 ka BP, where the $\delta^{13}C_{spel}$ decreases from ~-1.7 to -7.8‰ VPDB. Importantly, the absolute values and the magnitude of changes in $\delta^{13}C_{spel}$ in all three stalagmites are comparable over the study period.

The DCF is relatively low in the younger part of the record (~16-4 ka BP) of all three stalagmites (averages of 6.7%, 7.2%, 13% for Candela, Galia, and Laura, respectively, Fig. 3). DCF in Candela is slightly higher in the LG portion of the record (~11-15%, 18-20 ka BP), while values during the LGM (24 ka BP) are comparable with the EH. We disregard the one negative (and physically impossible) DCF value at 24.9 ka BP, as this is probably an artefact due to issues with U-Th dating in this section (open-system conditions in the basal section of Candela and potentially instrumental issues). For the modelling we use

a value of 7%, which is similar to values obtained for nearby paired U-Th – $^{14}$C samples (e.g., 24.2 ka BP and 24 ka BP; Suppl. Table 2). The DCF in the LGM sample from Galia is much higher (23%) than any in the three stalagmites, but there is no indication for alteration or other reasons why this sample should not be trusted.

While the absolute $\delta^{44/40}Ca$ values in the individual stalagmites are very different, probably reflecting variations in drip path length and drip interval, leading to different amounts of PCP, their temporal variation is remarkably small. In Candela, a slight

tendency towards less negative $\delta^{44/40}Ca$ values can be observed during H1, while values are lower during the LGM, the YD, and in the EH (Fig. 3). $\delta^{44/40}Ca$ values in Galia and Laura are within uncertainty of each other. The $\delta^{44/40}Ca$ values of the three bedrock samples are consistent, suggesting a homogeneous source of Ca for the three stalagmites (Fig. 3). This allows us to calculate $f_{Ca}$ and quantitatively estimate the amount of PCP for the stalagmites. By their nature, $f_{Ca}$ values mirror the $\delta^{44/40}Ca$, and suggest that Galia was subject to PCP to a much higher degree than Candela and Laura, where $f_{Ca}$ is comparable. As for

the $\delta^{44/40}Ca$, $f_{Ca}$ values in all three stalagmites indicate no major changes over the deglaciation, suggesting minimal changes in PCP.

Comparing the three proxies to temperature reconstructions from the Iberian Margin (Darfeuil et al., 2016), using linear interpolation to roughly match the different records, confirms a negative correlation between $\delta^{13}C_{spel}$ and temperature (-0.63 to -0.9‰ °C$^{-1}$, r$^2$ = 0.67 – 0.96), while the relationship between $\delta^{44/40}Ca$ and DCF to temperature is weak and/or inconsistent

(Fig. 4).

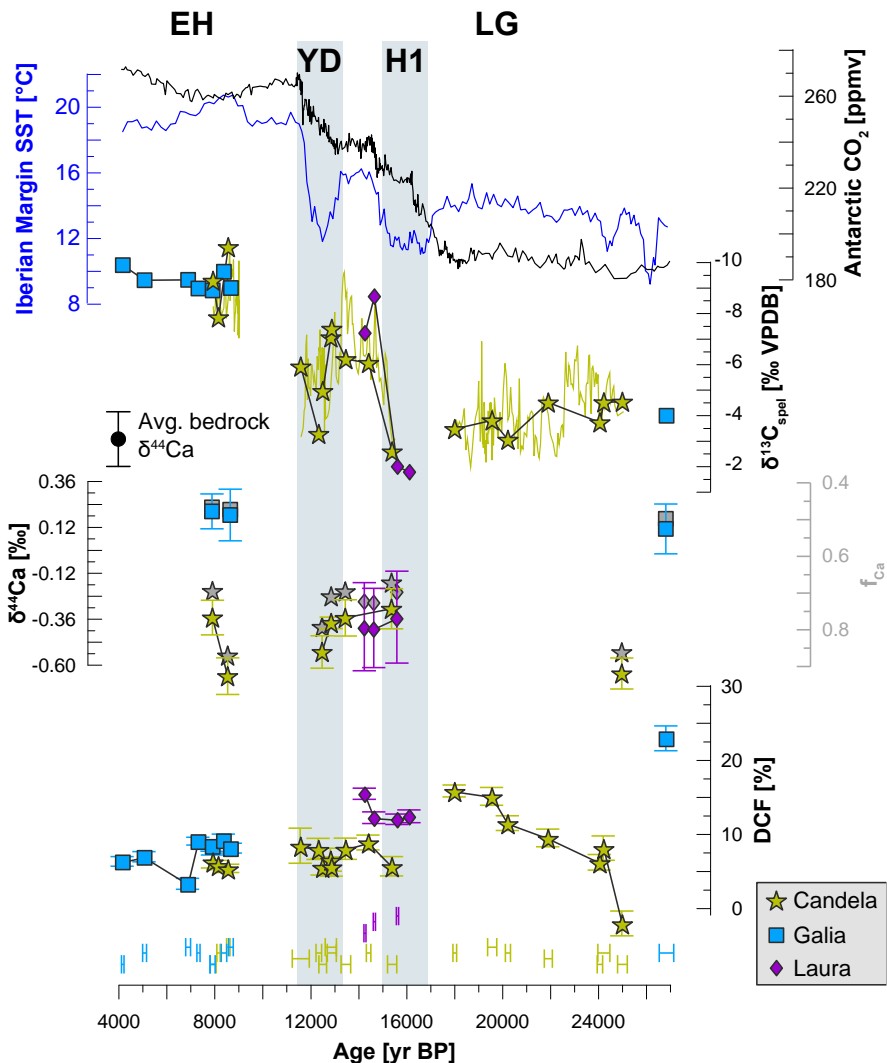

Figure 3: Proxy records from stalagmites Candela, Galia, and Laura over time, compared to regional temperature reconstructions (TEX$_{86}$-derived sea surface temperatures from the Iberian Margin; Darfeuil et al., 2016) and global $CO_2$ (ice core composite from Antarctica; Bereiter et al., 2015). The high resolution $\delta^{13}C_{spel}$ record from Candela (thin green line) is shown for reference and was originally published in Moreno et al. (2010). The time periods (LG, EH) at the top of the figure indicate the intervals used for the modelling to define temperature and atmospheric $pCO_2$.





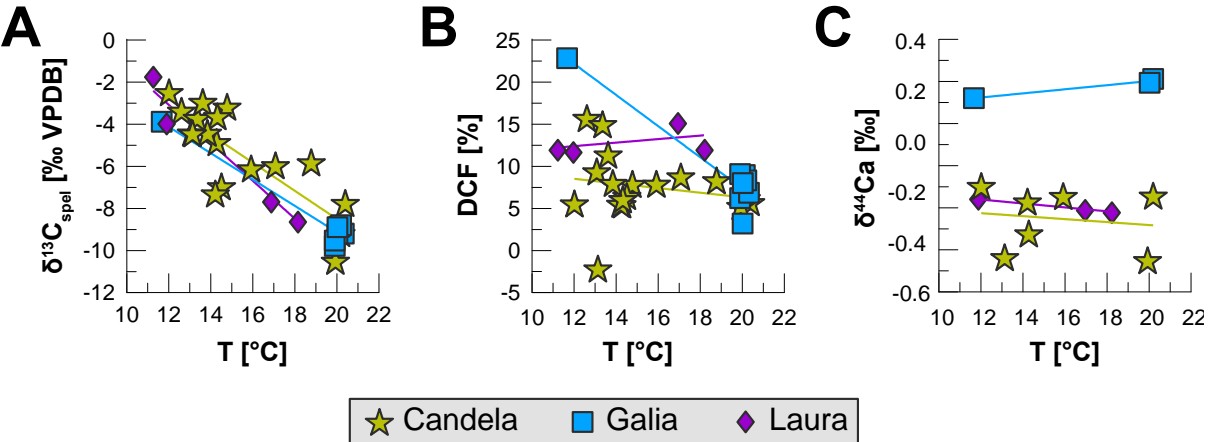

Figure 4: Stalagmite proxies vs. temperature, colour-coded by stalagmite. A – $\delta^{13}C_{spel}$, B – DCF, C – $\delta^{44/40}Ca$. The
corresponding palaeo-temperatures are linearly interpolated from the Iberian Margin SST record by Darfeuil et al. (2016). Of
the three proxies, $\delta^{13}C_{spel}$ shows the strongest and most consistent relationship to temperature, while the other proxies show
weak or inconsistent relationships.

**4.2 Modelling**

Each mixing line produced 5940 solutions for the LG and EH scenarios, respectively (396 for each combination of soil $pCO_2$
and $\delta^{13}C$). However, only about 41% (LG: 2457, EH: 2445) of the simulations resulted in carbonate precipitation, while for
the rest, precipitation was inhibited by the solution not reaching supersaturation with respect to calcium carbonate.
Supersaturation was not reached where low soil $pCO_2$ or closed system conditions reduced the amount of carbonate being
dissolved, or where the difference between cave air $pCO_2$ and solution $pCO_2$ was very small or negative. Thus, there is no
need to further prescribe the cave $pCO_2$ as a fraction of the soil $pCO_2$, as simulations with unrealistic parameter combinations
(i.e., higher cave $pCO_2$ than soil $pCO_2$) are automatically discarded.

Simulations from all three mixing lines produce results that match the stalagmite DCF, $\delta^{44/40}Ca$ and $\delta^{13}C_{spel}$ within measurement
uncertainty (Fig. 5). Thus, the initial parameter selection was sufficient to constrain the system and the estimate of the soil
respired end member composition is accurate. Test simulations using a respired end member with $pCO_2$ higher than 7800
ppmv consistently lead to overestimation of stalagmite $\delta^{44/40}Ca$ values, further validating the initial parameter selection.

The matching solutions from mixing line 1, which most closely reflects conditions at the cave sites, show an increasing trend
in median soil $pCO_2$ values over the deglaciation (Fig. 5). Soil $pCO_2$ values are consistently lower during colder time periods
(LGM, H1, and YD) and increase during warmer periods (BA and EH), maximising with the onset of the Holocene. Cave
$pCO_2$ values show a similar overall trend of increasing values towards the EH and the system is characterised by semi-open to
open conditions (100 – 500 L gas per L of solution, Suppl. Figure 1). Increasing soil $pCO_2$ values over the last deglaciation





are also needed when using mixing lines 2 and 3. For mixing line 3, no matching solutions are found during the LGM and YD, a consequence of the more negative soil carbon end member $\delta^{13}C$ used (-25.91‰ VPDB).

The sensitivity analysis allows more degrees of freedom in the model, where soil $pCO_2$, soil $F^{14}C$, cave $pCO_2$, and gas volume are allowed to freely vary but soil gas $\delta^{13}C$ is kept constant at -18‰. While solutions matching DCF and $\delta^{44/40}Ca$ are easily

305    found with this set of parameters, the deglacial trend in $\delta^{13}C_{spel}$ cannot be reproduced (Fig. 5). Only ~2‰ of the ~6‰ decrease in $\delta^{13}C_{spel}$ can be explained through processes other than changes in the soil gas $\delta^{13}C$ (Fig. 6). However, about half of the decrease in $\delta^{13}C_{spel}$ between H1 and the BA can be explained without invoking changes in soil gas $\delta^{13}C$. It should be noted that the absolute value of the residual calculated from the $\delta^{13}C$ is tied to the initial parameter selection, and would vary if we chose differently. The relative differences however, would remain the same, as long as the initial soil gas $\delta^{13}C$ is not allowed to vary.

310    We have chosen a relatively high initial soil gas $\delta^{13}C$ (-18‰) as more negative values result in very few solutions matching the proxy data.







Figure 5: Modelling results compared to measured proxies in stalagmite Candela. Stalagmite measurements ($\delta^{13}C_{spel}$, DCF, $\delta^{44/40}Ca$; black dots) are compared to best fitting model solutions (colour-coded by simulation type). Simulation results are shown as box plots, with the median and upper and lower quartiles displayed. Outliers are shown as coloured dots. Grey shading indicates intervals of the measured proxy values used to filter the simulations. The soil $pCO_2$ derived from the different model solutions is shown. The time periods (LG, EH) at the top of the figure indicate the intervals used for the modelling to define temperature and atmospheric $pCO_2$.

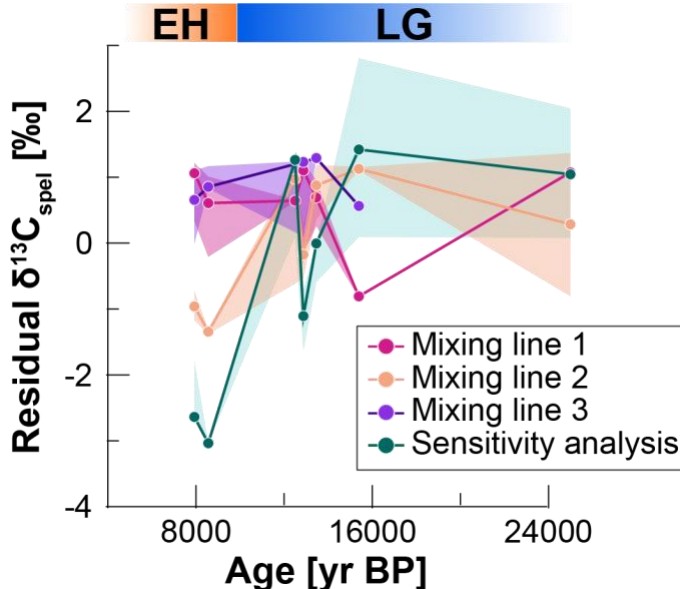

Figure 6: Residual $\delta^{13}C_{spel}$ calculated as the difference between measured and modelled $\delta^{13}C_{spel}$ over time.

## 5 Discussion

### 5.1 Temperature sensitivity of soil respiration as main driver for $\delta^{13}C_{spel}$

Combined multi-proxy analysis on three stalagmites and geochemical modelling provide strong evidence that changes in initial soil gas $\delta^{13}C$ are necessary to explain the deglacial trend in $\delta^{13}C_{spel}$ observed in northern Spain. Here we show that this trend is best explained by variations in soil respiration and in the relative proportion of respired vs. atmospheric $CO_2$ in soil gas. Soil gas is a mixture of $CO_2$ produced by respiration and atmospheric air (Amundson et al., 1998). Therefore, the $pCO_2$ and isotopic composition of soil gas over depth can be modelled by a mixing line between the atmospheric and soil carbon end member (Pataki et al., 2003). While more recent research has pointed out that this approach neglects spatio-temporal fluctuations in the isotopic signature of soil $CO_2$ sources (Goffin et al., 2014), as well as soil storage capacity and the possibility of turbulent transport (Maier et al., 2010), it still provides a valid model with which we can test the overall effects of bulk variations in soil





respiration on the dripwater solution. Our modelling results show a consistent pattern of increasing soil $pCO_2$ over the last deglaciation, with absolute values ranging between ~530-1030 ppmv during the LGM, and ~1155-5780 ppmv during the EH. An increase in soil respiration rates coinciding with Holocene warming is likely, as higher temperatures promote more rapid

soil carbon turnover (Vaughn and Torn, 2019) and the establishment of denser forests (Vargas and Allen, 2008). Climate model simulations confirm that net primary productivity in northern Spain was lower during the LGM than at present (Scheff et al., 2017). Pollen studies from northern Spain show significant and rapid changes in vegetation type and cover over the Pleistocene-Holocene transition (Moreno et al., 2014). While LG pollen reconstructions suggest a landscape dominated by open grassland (30-35% *Poaceae*) with significant steppic taxa and low arboreal pollen (30-50% primarily *Pinus sylvestris* and *Betula*), the

EH pollen assemblage is dominated by arboreal pollen (70-90%; Moreno et al., 2011). It is likely that the rapid response of pollen assemblages to climate warming is due to the region's proximity to documented tree refugia in the Mediterranean region (Fletcher et al., 2010).

Assuming a temperature change of roughly 8°C between the LGM and EH (Darfeuil et al., 2016), the sensitivity of soil respiration to temperature change ($Q_{10}$, i.e., factor by which soil respiration increases with a 10°C rise in temperature) derived

by our modelling experiments lies between 2.7 and 7, depending on the initial conditions of the models. This tends to be higher than the mean global $Q_{10}$ values of $3.0 \pm 1.1$ found by the soil respiration database (Bond-Lamberty and Thomson, 2010), and may highlight an additional contribution from changing substrate to the initial soil gas $\delta^{13}C$ over time, which would alter the $\delta^{13}C$ of the respired soil end member itself (Boström et al., 2007). We can exclude changes in vegetation assemblage from C4 to C3 plants, as there is no evidence for widespread presence of C4 plants during the glacial in Northern Spain (Moreno et al.,

2010) or elsewhere at temperate Western European sites (Denniston et al., 2018; Genty et al., 2006, 2003). A change in the balance between heterotrophic and autotrophic respiration is another possibility that would influence the soil gas $\delta^{13}C$. Changes in temperature affect root and microbial respiration differently (Wang et al., 2014), as do changes in other environmental variables, e.g., precipitation regimes and nutrient cycling (Li et al., 2018). Microorganisms are typically enriched by 2-4‰ compared to plants (Gleixner et al., 1993), and vertical enrichment by ~2.5‰ in soil profiles has been attributed to an increasing

contribution of soil microbially derived material with depth to the overall soil carbon turnover (Boström et al., 2007). The release of older and enriched carbon from soils and long-lived plant material through respiration could provide an additional mechanism with which the soil gas $\delta^{13}C$ could be shifted regardless of changes in soil respiration (Fung et al., 1997). Given the high $Q_{10}$ values obtained by our model results, it is likely that some of the shift in soil gas $\delta^{13}C$ is related to a change from a more enriched to a more depleted substrate over the deglaciation and/or an increase in photosynthesis over respiration, as

might be expected with a proportional increase in vegetation cove. A higher respired $\delta^{13}C$ during the LG is also suggested by the model results, where mixing line 3 with the lowest respired $\delta^{13}C$ (-25.9‰) fails to produce solutions matching the speleothem data (Fig. 5). Given the small variation in DCF values in Candela over the deglaciation, we can exclude the possibility that changes in the fraction of bedrock carbon from changing dissolution conditions constitute an important driver of the deglacial signal.





Another intriguing possibility is that the carbon isotopic fractionation of C3 vegetation is controlled by atmospheric $pCO_2$ (Schubert and Jahren, 2015). A recent global compilation of speleothem records shows that, after correcting for the expected effect of precipitation and temperature on $\delta^{13}C$ of C3 biomass and the temperature-dependent fractionation between $CO_2$ and calcite, the global average $\delta^{13}C_{spel}$ closely tracks atmospheric $pCO_2$ over the last 90 ka (Breecker, 2017). The magnitude of the deglacial shift in C3 plant $\delta^{13}C$ has been proposed to lie around 2.1‰ (Schubert and Jahren, 2015). The deglacial $\delta^{13}C_{spel}$ record

from northern Spain however, shows clear millennial-scale variations that coincide with temperature variations, but are not driven by atmospheric $CO_2$ (Fig. 3). Therefore, while it is possible that a $CO_2$ fertilization effect contributed to the overall decrease in $\delta^{13}C_{spel}$ over the deglaciation, this effect is likely not dominant.

## 5.2 Other processes affecting $\delta^{13}C_{spel}$

While a change in soil respiration and consequently in the proportion of respired vs. atmospheric $CO_2$ in the soil gas can explain the deglacial trend in $\delta^{13}C_{spel}$, a number of other, cave-specific processes could also contribute to changes in $\delta^{13}C_{spel}$. The direct effect of the glacial-interglacial temperature change on carbonate equilibria and fractionation factors is small and taken into consideration by running the simulations with EH and LG parameters. It is more difficult to assess whether kinetic fractionation effects affected the stalagmite at different times, potentially amplifying the $\delta^{13}C_{spel}$ signal. CaveCalc uses standard

kinetic fractionation factors for the $CO_2$-DIC-carbonate system (Romanek et al., 1992; Zhang et al., 1995) and therefore such variations are not considered by the model. However, the high degree of coherence between $\delta^{13}C_{spel}$ records from the entire Western European region suggests that localised, cave-specific kinetic fractionation effects likely played a minor role in driving the deglacial trend (Fig. 1).

Changes in the amount of PCP the dripwater experiences *en route* to the speleothem can lead to significant variability in $\delta^{13}C_{spel}$

records (Fohlmeister et al., 2020), and are tightly coupled to changes in cave air $pCO_2$ and cave ventilation dynamics. Higher cave air $pCO_2$ and a reduced $CO_2$-gradient between the supersaturated dripwater solution and the cave air result in less PCP, and vice-versa for lower cave air $pCO_2$. It is likely that cave $pCO_2$ was lower during the last glacial at the study sites, and indeed this is also suggested by our model results (Suppl. Fig. 1). Cave $pCO_2$ is coupled to soil $pCO_2$, which provides its upper limit, and model results automatically filter out unrealistic scenarios, as no speleothem precipitation occurs when cave $pCO_2$

is equal to or higher than soil $pCO_2$. Our multi-proxy dataset allows us to evaluate the importance of PCP on $\delta^{13}C_{spel}$ quantitatively, as $\delta^{44/40}Ca$ can provide quantitative PCP reconstructions over time (Owen et al., 2016). Mg/Ca ratios are also often used as proxy for PCP, however, caution is required in their interpretation in Pindal Cave because in the Holocene, Mg/Ca is also affected by increasing surf-zone marine aerosol contributions as rising sea level brought the coastline to the foot of the sea cliff in which the cave has its entrance (Suppl. Fig. 2 and 3). Over the last deglaciation, $\delta^{44/40}Ca$ and $f_{Ca}$ varied only

minimally in both Candela and Galia (Fig. 3), suggesting that changes in PCP were small. This is also reflected in the sensitivity analysis, where changes in $\delta^{13}C_{spel}$ cannot be reproduced while also fitting the $\delta^{44/40}Ca$ curve (Fig. 5). CaveCalc uses cave $pCO_2$ to match the degree to which dripwater has lost its initial Ca due to calcite precipitation, giving us a measure for PCP.



A solution equilibrated with a high soil $pCO_2$ would lose the majority of its carbonate in a simulation where cave $pCO_2$ is atmospheric, due to the high degree of oversaturation of the dripwater solution compared to cave air. If $\delta^{44/40}Ca$ provides

evidence that only a small portion of Ca has been precipitated, then the simulation must match the data by prescribing a higher cave $pCO_2$. In reality, the fraction of Ca precipitated from dripwaters depends not only on the oversaturation of the solution, but also on the time the water is present as a thin film on the cave ceiling and stalagmite surface before being replaced by a new water parcel (i.e., drip interval; Fohlmeister et al., 2020; Stoll et al., 2012). When the drip interval is short, each water parcel won't have enough time to fully degas $CO_2$ and equilibrate with the cave atmosphere, and the actual PCP is lower than

what would be possible given the cave $pCO_2$. CaveCalc does not model drip interval, and therefore the cave $pCO_2$ inferred from the simulations might be overestimated. We test the effect of drip interval length changes on $f_{Ca}$ and PCP using the forward model ISTAL (Stoll et al., 2012), which explicitly models this parameter. Two model scenarios mimick full glacial and Holocene conditions, including changes in temperature, cave $pCO_2$, and soil $pCO_2$ for "winter" (i.e., atmospheric) and "summer" (i.e., elevated) cave $pCO_2$ (Suppl. Fig. 4). The effect of the glacial-interglacial temperature change is only significant

for high drip intervals during the cold season, where PCP is slightly higher during interglacial conditions. At high drip intervals, the temperature increase leads to a change in $f_{Ca}$ of ~-0.1, which translates to a ~0.04‰ change in $\delta^{44/40}Ca$ and a -0.7‰ VPDB change in $\delta^{13}C_{spel}$. This corroborates our expectation from the $\delta^{44/40}Ca$ record and CaveCalc model results, suggesting that only a small part of the shift in Candela $\delta^{13}C_{spel}$ over the last deglaciation was due to changes in PCP.

While it is likely that some or all of these processes affected the deglacial $\delta^{13}C_{spel}$ to some extent, their magnitude is not large

enough to explain the measured ~6‰ shift, suggesting that changes in soil $pCO_2$ played a significant role.

**5.3 Insights into regional hydroclimate over the last deglaciation**

Our new multi-proxy record from stalagmites from northern Spain also offers nuanced insights into local hydroclimate conditions over the last deglaciation. While DCF mainly responds to changes in carbonate dissolution conditions, and therefore

is sensitive to changes in infiltration, $\delta^{44/40}Ca$ is driven by both infiltration dynamics (determining the initial oversaturation of dripwater and the degassing timescale) and cave atmospheric $pCO_2$ (determining the amount of PCP occurring). The Candela record suggests no substantial shift in infiltration dynamics or PCP occurring between LG and EH (Fig. 3), as both proxies fluctuate around a mean value without long-term trends. This result suggests that the glacial hydroclimate was not significantly different from the Holocene, and stands at odds with previous mainly pollen-based studies that often point towards a drier

glacial, but with considerable variability over millennial timescales (Fletcher et al., 2010). Recent modelling results have challenged the interpretation of the glacial being cold and dry, suggesting instead that, while precipitation was lower during the LGM, topsoil moisture was actually higher than at present (Scheff et al., 2017). Our new stalagmite data supports this interpretation, suggesting that temperature, and not hydroclimate conditions, were the main drivers of ecosystem productivity over the deglaciation.




## 6 Conclusions

We have combined multi-proxy ($\delta^{13}$C, $\delta^{44/40}$Ca, and DCF) data from three speleothems and quantitative geochemical modelling to show that the temperature sensitivity of $\delta^{13}C_{spel}$ over the last deglaciation in Western Europe is best explained by c. Generating a large ensemble of forward models of processes in soil, karst, and cave allows estimation of their likely importance

and variability over time. Speleothem geochemical proxies that are sensitive to different components of the soil-karst-cave system can be employed to extract the most likely model solutions from the ensembles, and thus quantifying the system's initial conditions, particularly soil $pCO_2$. Our approach involved the coupling of soil $pCO_2$ and soil $\delta^{13}$C values, as expected when following a mixing line between a soil and an atmospheric end member, and thus allowing us to model changes in soil respiration. While uncertainties remain, in particular with respect to possible changes in the soil carbon end member over time,

we find that an increase in soil respiration is necessary to explain the large shifts in $\delta^{13}C_{spel}$ over the last deglaciation in Spain. Given the exceptional regional coherency of $\delta^{13}C_{spel}$ records over temperate Western Europe, it is likely that this effect is of broader regional significance. Our study is the first to quantitatively model environmental processes in karst systems using a multi-proxy approach, and paves the way towards more nuanced interpretations of $\delta^{13}C_{spel}$ records. Moreover, our multi-proxy records support recent climate model results that reject the long-standing "drier and colder glacial" notion in Western Europe,

pointing instead toward a dominant forcing of temperature on ecosystem productivity, rather than hydroclimate.

### Acknowledgements

This study was funded by the Swiss National Science Foundation (SNSF) grant P400P2_180789 awarded to F. Lechleitner, by ETH core funding to H. Stoll, and by doctoral Fellowship ETH-13 18-1 to O. Kost. We thank Yu-Te (Alan) Hsieh for

assistance with the $\delta^{44/40}$Ca measurements at the University of Oxford, and Madalina Jaggi at ETH Zurich for measurements of $\delta^{13}$C and trace elements, and Saul Gonzalez-Lemos for cave air sampling.

### Data availability

The code used for calculation of the stalagmite dead carbon fraction can be found at

(https://github.com/flechleitner/DCF_calculator). All data used in the study and codes for the modelling can be found at https://github.com/flechleitner/Spain_analysis and in the supplementary information provided with the article.

### Author contributions

F. Lechleitner, H. Stoll, and G. Henderson designed the study and acquired funding for the project. F. Lechleitner, N.

Haghipour, and C. Day performed the geochemical analysis on speleothem samples. O. Kost collected and measured cave air samples from La Vallina Cave and aquired funding for the monitoring work. F. Lechleitner and M. Wilhelm performed the modelling experiments in CaveCalc and wrote the R code for the data-model evaluation. F. Lechleitner wrote the manuscript





and generated the figures. H. Stoll and C. Day provided additional input to the text. All authors provided feedback to the manuscript and approved it before submission.


**Competing interests**

The authors declare that they have no conflict of interest.

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
