# Peer review of "Stalagmite carbon isotopes suggest deglacial increase in soil respiration in Western Europe driven by temperature change"

_Climate of the Past, 2021_

## Author Comment (AC1)

**Response to anonymous referee #1**

This paper presents a useful modeling approach for reconstructing soil respiration using stalagmite carbon isotopes and proxy constraints on prior calcite precipitation and bedrock dissolution effects. Carbon isotopes have been a particularly messy avenue in speleothem science due to the complex interplay of these effects (and others) and the paper represents an exciting effort toward rigorously disentangling this mess. The model is an important step toward understanding how soil respiration changes with climate and, given the breadth of data available in the SISAL database, could be quickly applied on a large scale (assuming it can be appropriately constrained). But the modeling approach carries some critical (and likely invalid) assumptions that need to be addressed. **This paper can be of sufficient interest for Climate of the Past and I think my points can be addressed with major revisions**. I commend the authors for their coupled proxy-model approach and hope my feedback is useful as they refine their work.

My expertise most closely aligns with the modeling work, so I focus my feedback on this part of the manuscript. I can't speak much to the analytical methods. Below, I've divided my feedback into points about the main modeling approach (outlining my own confusion) and line-by-line items. My biggest concerns are that the modern calibration of the soil respired end member seems invalid (or at least should use modern CO2 levels), and that it is not a safe assumption that the mixing end members are time-invariant.

Response: We thank the reviewer for their thorough and fair assessment of our manuscript. We address the comments in detail below.

**Main modeling points**
My understanding of the modeling approach involves three steps: (1) use modern data to derive a relationship between atmospheric CO2 and soil-respired end members; (2) assuming this mixing relationship holds through time prescribe the full range of soil CO2 and $\delta^{13}C$ possibilities to solve for proxy data with CaveCalc; (3) using the model output, select and analyze the combination of input parameters that yield results closely in line with the measured data. Below, I dive into my concerns on steps 1 and 2 in more detail and include one note on step 3.

**Step 1: calibrating a soil respiration end member**
This calibration exercise (outlined in Figure 2) carries four big assumptions that I think need to be addressed. The biggest has to do with using modern data to calibrate a pre-industrial mixing curve (see point 2).
1.  First, cave-monitored CO2 and $\delta^{13}C$ are used to calibrate a soil CO2 mixing line. This assumes that the mixing of atmosphere and respired CO2 in the cave falls the exact functional form of mixing in the soil (put otherwise, it assumes that the bedrock contribution to cave carbon is the same as soil carbon and that there are no other carbon fluxes distinguishing cave from soil). The authors concede this does not hold true in the winter, but could more be said about this assumption in the months of April-November (when (I think) the monitoring data are used)? Could changes in hydrology, or cave vs soil temperatures, or other things violate this assumption? Is soil CO2 assumed to reflect the soil-column integrated conditions? Is it a problem that this assumption breaks down seasonally if calcite deposition is seasonally biased? **I think, at the least, it must be written that this assumption is made** (right now the link between cave conditions and soil conditions is a bit vague to me).

Response: We agree that the assumptions made when calibrating the soil gas end member would benefit from clarification in the text. Here we summarize our reasoning on using cave air measurements to constrain soil gas and estimate the respired end member. We now show the actual monitored data with modern atmospheric composition and post industrial mixing line, as well as our suggested pre-industrial mixing line corrected for the Suess effect. We will add relevant clarifications to the manuscript.

**Estimation of soil respired end member from cave $CO_2$ measurements:** In the absence of a full soil monitoring campaign, samples collected from the cave in summer months represent a reasonable approach to estimate the isotopic value of the respired end member contributing to soils/epikarst (Figure 1). This is because the cave, like the soil, is defined by a two-end member mixing system, which is driven by the physical ventilation of the cave. The main fluxes of carbon in a system like El Pindal and La Vallina caves are from soil gas (mainly seeping through the host rock and into the cave) and atmospheric air (through ventilation). Like many mid-and high latitude cave systems, there is a seasonal reversal in the airflow direction in La Vallina cave (Stoll et al., 2012). In the summer, when cave air is colder than exterior air, cave air flows out the entrance, and is replaced by inflow and diffusion of soil/epikarst air. In this season, the cave has the highest $CO_2$ concentrations and points which fall closer to the soil respired end member on the Keeling plot. In the winter, when cave air is warmer than exterior air, exterior air flows in through the cave entrance, bringing the cave closer to the atmospheric end member.

The data from the monitoring of the cave reflects primarily $CO_2$ from the soil that is drawn through the karst network into the cave. It is therefore likely reflecting soil column-integrated conditions and the full contribution of respired $CO_2$ in the soil and epikarst unsaturated zone (below the soil, "ground air").

Any contribution of C flux from bedrock dissolution does not significantly affect our estimation of the respired end member, because the intercept defining the respired end member is most influenced by the summer season cave $pCO_2$ data. During the summer season drip flow rates are more than an order of magnitude lower than in the winter and degassing from this drip is suppressed by the high cave $pCO_2$. In winter, when drip rates are higher, and cave $pCO_2$ is lower, then degassing may contribute to C in cave air, as seen in other systems (Waring et al., 2017). However, winter data corresponding to ventilated periods are near to the atmospheric composition, and we explicitly define our atmospheric end member from global measurements, not the local cave measurements. Hence, the ventilated season measurements, which in theory may have contribution from degassing of dissolved limestone, would have an insignificant impact on our calculated mixing line. Furthermore we do not find evidence for a different intercept in winter and summer end members, unlike monitoring studies which infer a strong effect of degassing of a carbon source from limestone dissolution (Waring et al., 2017). (We also note that contribution of C to stalagmites from bedrock dissolution is explicitly accounted for by modeling of the dead carbon fraction, DCF, in stalagmite data in CaveCalc.)

We therefore argue that summer cave air can be used to estimate the isotopic composition of the respired end member of soil $CO_2$, when the competing fluxes are minimized: ventilation is reduced during the summer months (Stoll et al., 2012) and the higher cave $pCO_2$ levels reduce the amount of degassing that can occur from dripwaters entering the cave (reducing the contribution from host rock carbon).

An estimation of the modern respired end member, defined along a mixing line which includes the modern global atmospheric end member, is -26.9 +/- 0.8 ‰.

[Figure]

*Figure 1: Keeling plot of cave monitoring data from La Vallina Cave. The regression was calculated using the full dataset and the global atmosphere (purple dot). No clear seasonal bias in sample composition stands out.*

**Seasonally biased calcite deposition**
The seasonality of the modern cave air has the advantage of helping to define the modern respired end member. From our monitoring, we do not find evidence for a different isotopic value of the respired end member in different seasons. Thus, exploiting the seasonal ventilation of the cave to define the mixing line and respired end member does not preclude using this respired end member to interpret records from speleothems in which deposition is dominant in one season. Monthly monitoring of drip rate, dripwater chemistry, and cave air composition indicates that currently in the cave, some stalagmites grow at similar rates throughout the year, whereas in other sectors of the cave, growth occurs exclusively in winter, driven by cave ventilation.
We will discuss this in the next version of the manuscript.

2. Second, the model is calibrated to pre-industrial CO2 levels even though the data are taken in modern conditions (when CO2 is very well-constrained and much higher!) I think this is done so the same calibration end members can be applied throughout the Holocene (more on this in the "Step 2" section). The forest data are ignored because they might be influenced by "turbulence and advection effects" (Line 196),ut they would probably fit really nicely on a mixing line that reaches to a modern CO2 end member (with higher CO2)! In fact, all of the data would likely fit better on such a line (given that the monitoring data residuals to mixing line 1 are mostly above the line when CO2 > 1,000 ppm). This makes sense, because the monitored data are mixing with the modern atmosphere, not the pre-industrial

atmosphere. (Correcting for the Suess effect only corrects for pre-industrial $\delta^{13}C$, it does not account for the difference in CO2 between then and now). **I highly encourage the authors to use a calibration to a modern end member**. Otherwise a much more rigorous justification for the pre-industrial end member is needed.

Response: As suggested by the reviewer, we agree that it is useful to illustrate the full derivation of the modern, post-industrial respired end member also accounting for the modern atmospheric contribution, as a first step (Figure 1). As described in our response to point 1, including the modern atmospheric end member leads to an estimated respired end member of -26.9 ‰. This end member may be more negative than the preindustrial end member (which characterized the Early and mid Holocene growth periods of the stalagmites in this study), because atmospheric $\delta^{13}C$ has decreased by 2 ‰ over the last century due to anthropogenic activities (Suess effect). If the currently respired end member is composed of modern respiration and respiration of young (decadal age) soil carbon pools, the pre-industrial respired end member may have been as much as 2 ‰ heavier, that is closer to -25 ‰. If a significant fraction of the respired pool is older, then the preindustrial respired end member may fall between -27 and -25 ‰. We favour the less negative estimate because a young age of respired carbon is suggested by the rapid post-bomb spike decrease in $^{14}C$ in actively growing stalagmites in the cave.

Therefore, we define an updated reference mixing line for the Early Holocene as having a respired end member of -25 ‰ and an atmospheric end member of 260 ppmv and -6.3 ‰, consistent with ice core records. We use a pre-industrial atmospheric composition in the model since this more closely reflects the end member at the time of stalagmite growth.

3. Third, I don't know how the soil-respired end member is defined as 7800ppm and -22.9‰. **I imagine that it's an extrapolation of mixing line 1, but why not extrapolate some number other than 7800?** Is there some assumption that I'm missing?

Response: We have clarified that we define the isotopic value of the respired end member on the basis of Keeling plots, however by definition the Keeling intercept is not associated with a particular soil $CO_2$ concentration. We clarify that 7800 ppm $CO_2$ (now updated to 8000 ppmv, see step 2) is the soil concentration which best simulates the observed speleothem growth rates and isotopic ratios and we note it is consistent with soil $pCO_2$ above other caves in comparable settings (e.g. Borsato et al., 2015).

4. Fourth, **I think this modeling approach assumes that boundary layer CO2 concentration and $\delta^{13}C$ (the stuff that diffuses into the soil) also falls on the same mixing curve**. This should be stated since at least two things relevant to this study might violate this assumption. First, a shift from no canopy in the last glacial to a canopy when forests appear might lead to a "canopy effect" whereby $\delta^{13}C$ gets lower than expected for a given pCO2 due to recycling. Second, "turbulence and advection effects" (line 196) that appear to matter during the daytime (probably when photosynthesis is happening) can overprint the simple mixing relationship and propagate down to the soil respired end member. These effects might well be small, but I think the assumption should at least be recognized.

Response: This is a good point, and we will add this to the discussion of the uncertainties in the model.

*Step 2: assuming this holds through time*

This analysis assumes that the mixing slope between soil respired $\delta^{13}$C and atmospheric $\delta^{13}$C is constant through time. This is not a good assumption because a lot of factors that matter on decadal or longer timescales (i.e. factors that are not captured by the short calibration) violate it by changing end member CO2 but not $\delta^{13}$C (or vice versa). For example, if CO2 increases (like it did from LG to EH), then the isotopic composition of CO2 must decrease to keep the end member on the curve, but we know very well that this assumption is violated on paleoclimate timescales (e.g. Schmitt et al., 2012; Science; Figure 1). Similarly, if soil respiration decreases (thus decreasing soil-respired CO2) then the $\delta^{13}$C of soil respiration must increase to stay on the mixing curve. I'm not sure if there's a defensible mechanism for this, although it might occur by coincidence if water stress increases vegetation $\delta^{13}$C (thus soil-respired $\delta^{13}$C) while decreasing soil respiration. Either way, I am not aware of any mechanistic reason why the end members of the mixing relationship should, themselves, vary along a mixing curve. The end-members, just like the average soil CO2 values that reflect their mixing, should vary over time.

Response: Two factors may contribute to variation in the mixing line over time. The first and most certain factor is due to changes in the concentration of atmospheric $CO_2$. The isotopic composition of the atmospheric $CO_2$ remains within a few tenths of a permil of the Holocene value (Schmitt et al., 2012). Therefore, assuming a constant respired end member, the slope of the mixing line is reduced during periods of lower atmospheric $pCO_2$ (Figure 2). As a consequence, for a given soil $pCO_2$ value, the isotopic composition of soil $pCO_2$ is more negative during the glacial than during the interglacial. This effect is very small (<0.4 ‰) for soil $pCO_2$ 4000 ppm or higher, but for soil ppmv of 2000 ppm corresponds to a 0.9 ‰ difference and at 1000 ppm corresponds to a -1.8 ppmv difference. Nonetheless, this is still a small effect compared to the range of $\delta^{13}$C resulting from changes in the soil $pCO_2$. If anything, this change in the mixing line would attenuate the difference in $\delta^{13}$C between a low $CO_2$ glacial soil and a high $CO_2$ interglacial soil.

[Figure]

*Figure 2: Left panel: Mean respired $\delta^{13}C$ of soil $CO_2$ across different biomes, adapted from Pataki et al., 2003. The respired end member used in this study is shown by the brown square (with uncertainty of +/- 3 ‰). Right panel: Mixing lines for soil gas illustrating expected changes in the atmospheric composition over glacial-interglacial transitions. While the isotopic composition remains virtually the same, the pCO₂ of the atmosphere varies more strongly, resulting in changes in the slope of the mixing line.*

The mixing line may also vary if the respired end member changes.  However, there are few constraints on potential changes in the end member.  Although there is variation in the respired end member both within and among biomes, the mean respired end member for the potential biomes which may have characterized this site over the last 25 ka  - temperate broadleaf, temperate conifer, and boreal - feature mean $\delta^{13}C$ of respired end members which differ by only 1 ‰  (Figure 2; Pataki et al., 2003).  This suggests that we cannot predict a systematic change in the $\delta^{13}C$ of the respired end member with changes in the biome.  Moreover, the fact that speleothems across Western Europe show very similar trends in $\delta^{13}C$ over the deglaciation also indicates that highly localised factors that may lead to a strong change in respired $\delta^{13}C$ without a biome change are unlikely.  Consequently, we address the potential for variation in the respired end member by completing a sensitivity analysis of mixing lines which encompass 3 ‰ heavier and lighter respired end members.

I think the modeling can still be performed if some significant changes are made (these are just suggestions and other options can be valid too!).
1.  Consider using the actual paleoclimate constraints on pCO2 and $\delta^{13}C$ of CO2 to parameterize the atmospheric end member.
Response: We have now updated the atmospheric end member for multiple time windows including the EH, LGM, and one period during the deglaciation, in accordance with Schmitt et al. (2012). These do not change the conclusions of the study because

they lie largely within the sensitivity range of the different mixing lines and are therefore taken into account (Figure 3).

[Figure]

Figure 3: Updated modelling results for stalagmite Candela using newly calculated mixing lines that consider changes in the atmospheric end member (EH, deglacial, and LGM), and in the respired end member $δ^{13}C$. All mixing lines were grouped together and compared to the sensitivity analysis keeping soil gas $δ^{13}C$ constant.

2. Instead of calibrating the soil-respired end member with modern data, be clear that constraints on this term are not great but define reasonable ranges and run sensitivity tests. Allow the soil respired $\delta^{13}$C and CO2 to vary with time. Or consider forcing the model with different scenarios as a sensitivity test (i.e. low vegetation $\delta^{13}$C, high vegetation $\delta^{13}$C and variable, or decreasing, or increasing $\delta^{13}$C). Given the strong evidence for substantial changes in vegetation, it is helpful (maybe necessary) to rule this out as the main factor affecting $\delta^{13}$C_spel. Consider holding soil respiration constant while letting $\delta^{13}$C -respired vary; one might find that the variability would have to be too high to be explained by changes in C3 vegetation or water stress alone (see Kohn, 2010; PNAS).

Response: We have explored a wide range of sensitivity in the composition of the respired end member, which we infer to be the suggestion of the reviewer (holding soil respiration constant while letting $\delta^{13}$C-respired vary) since the mixing line with the atmosphere is an ubiquitous feature of soil-epikarst system.

To test the sensitivity of the model to changes in the soil $pCO_2$, we have performed two additional analyses:
1) We have extended the new mixing line 1 (respired $\delta^{13}$C -25 ‰) up to $pCO_2$ of 15,000 ppmv (Figure 3). This allows us to test how the system reacts to coupled changes in $pCO_2$ and $\delta^{13}$C of the soil gas. Since at high $pCO_2$, the soil gas $\delta^{13}$C becomes insensitive to changes (hyperbole), this extrapolation mostly affects the $pCO_2$ of the initial solution, while changes in initial $\delta^{13}$C are minimal. We find that the trend in increasing $pCO_2$ over the deglaciation remains robust, as higher initial soil $pCO_2$ does not lead to solutions matching the stalagmite data.

[Figure]

[Figure]

[Figure]

*Figure 3: Model results for new mixing line 1 with maximum initial soil $pCO_2$ of 8000 ppmv (as used in the study, left), and increasing the maximum $pCO_2$ to 15000 ppmv. Higher $pCO_2$ does not lead to more solutions matching the stalagmite data.*

2) We calculated two more mixing lines with different $pCO_2$ of the respired end member (10,000 and 4,000 ppmv; Figures 4 and 5) while keeping the respired $\delta^{13}$C constant. Again, these simulations show a robust increasing trend in soil $pCO_2$ over the deglaciation, while the absolute values of the median $pCO_2$ change.

[Figure]

*Figure 4: Example of soil air pCO₂ that leads to solutions matching the stalagmite data when calculating a new mixing line with respired pCO₂ of 4,000 ppmv.*

[Figure]

*Figure 5: Example of soil air pCO₂ that leads to solutions matching the stalagmite data when calculating a new mixing line with respired pCO₂ of 10,000 ppmv.*

This sensitivity analysis shows that, while our model cannot reconstruct absolute soil pCO₂ values, the general trend over the last deglaciation is robust.

3) Holding soil pCO₂ constant and letting soil δ¹³C vary leads to the entire 6‰ change in speleothem δ¹³C being driven by changes in the respired δ¹³C. This is unrealistic, as biome-level values of respired δ¹³C typically show little variation (e.g., Pataki et al., 2003), and therefore even a substantial deglacial transition from boreal to forested landscape would likely not lead to such a large shift in δ¹³C.

We also argue that substantial changes in hydroclimate are unlikely over the deglacial transition in northern Spain: this is shown by our δ⁴⁴Ca and DCF records (sensitive to infiltration and carbonate dissolution/reprecipitation dynamics), which do not show any

temporal trends. It is also supported by recent climate modelling results, which do not suggest regional aridity during the last glacial (Scheff et al., 2017).

*Step 3: filtering for best model results*
More discussion / sensitivity analysis should be done here. Were other options for finding the "best fit" considered? How does changing the thresholds for carbon and calcium isotope data affect the results? What happens if one uses a broader DCF threshold? If these decisions affect the results (or if they don't) it would be important to know.
Response: The model is not very sensitive to the choice of DCF threshold. We have tested increasing the DCF confidence intervals to +/- 3% and this did not lead to any meaningful change in the results. Changes in $\delta^{44}Ca$ however, are more important, and we had to increase the confidence interval from the uncertainty from the proxy measurement, as lower uncertainty led to the model not finding matching solutions for all three proxies. Of course, this could be circumvented by performing more simulations including more different parameter combinations, but this is out of scope for this paper as it would not lead to different conclusions.
The different sensitivities of the model to DCF and $\delta^{44}Ca$ illustrate how $\delta^{13}C$ is affected more strongly by changes in the processes influencing $\delta^{44}Ca$ (mainly PCP) than those influencing DCF (open-closed system carbonate dissolution dynamics).

*Smaller comments and line-by-line*
- Please clarify the use of "soil carbon" vs "soil-respired carbon". For example, line 199 states "The regression points toward a soil carbon end member…". Is this treated as just "soil carbon" in the modeling? Because the exercise seems to imply that the constraint is a "soil-respired carbon" end member. Line 190 also refers to the "soil carbon end member" but states it was constrained with data, not the modeled regression (which I think is accurate). Since soil respired carbon is defined as a component of soil carbon (line 182) this distinction is super important. It's still not fully clear to me how soil carbon vs soil respired carbon are treated in the model.
  Response: We apologise for having caused confusion here. There are indeed differences between the soil respired carbon end member and the values used in the mixing line.
  As discussed above, at our cave sites we can treat cave air $CO_2$ during summer months as a mixture between atmospheric, soil-respired $CO_2$. With the Keeling plot approach, we determine the $\delta^{13}C$ of the respired end member during pre-industrial periods to be -25‰ (see discussion in point 2).
  For the modelling, we then define the mixing line along this regression, starting from a soil $pCO_2$ value that is reasonable considering our monitoring data and data from other comparable sites. This leads to the reported modelling end member of 8000 ppmv with a $\delta^{13}C$ of -24‰. As mentioned, we have now performed additional analyses using different end member compositions to test the robustness of the approach, which all lead to very similar results with respect to the deglacial trend in stalagmite $\delta^{13}C$.
  We will clarify our use of the terms in the manuscript.

- I imagine that the soil respired end member of the mixing curve changes seasonally. If calcite formation is seasonally biased, could this affect the results? For example, are more model solutions at higher soil CO2 conditions possible when strictly summer-time inputs are used?
  Response: Our monitoring data do not provide any evidence of a seasonal change in the $\delta^{13}C$ of the respired end member on the Keeling plot, as highlighted above. We

have modeled a single soil $CO_2$ and not a full seasonal cycle, but as analyses with higher $pCO_2$ have shown, such changes will not lead to different conclusions to our study.  Updated monitoring information (Kost et al, in prep) indicates that many locations do not have a strong seasonal bias in calcite formation.

Line 36: Check out Bova et al., 2021 (Nature) for updated Holocene climate constraints.
Response: We thank the reviewer for this suggestion and will add the reference in the next version of the paper.

Line 49: I'm not convinced by Figure 1 that these records are "highly consistent in timing, amplitude, and absolute $\delta^{13}C$". I worry that the words "highly consistent" are overstating the data. Consider focusing on the main trends that are clearly robust, like the general shift to lower $\delta^{13}C$ values from 18ka to 6ka.
Figure 1B: Consider labeling the El Pindal and La Vallina sites.
Figure 1C: Is the straight blue line from ~18ka to 15ka just due to the fact that there are no data? It might be clearer to disconnect the timeseries lines whenever there is a sufficiently long duration of no data (maybe wherever there is ~500 years of no data or something).
Response: We have changed the wording of the figure description in the text. The sentence now reads: "Speleothem carbon isotope ($\delta^{13}C_{spel}$) records from the temperate region of Western Europe are often clearly correlated to regional temperature reconstructions during the last glacial (Genty et al., 2003) and the deglaciation (Baldini et al., 2015; Denniston et al., 2018; Genty et al., 2006; Moreno et al., 2010; Rossi et al., 2018; Verheyden et al., 2014) (Fig. 1), pointing towards a regionally coherent mechanism driving the response to the temperature increase".
Figure 1B: We have added a label to the El Pindal-La Vallina site on the map.
Figure 1C: This issue was raised by both reviewers. The record in question from Villars cave has very low resolution, but no hiatus was reported at that depth. We have added a sentence clarifying this issue in the figure caption.

Line 129: I don't know if CP allows citing papers in review, just adding it here as a note (although I assume that the authors have already confirmed that this reference is okay!)
Response: Thank you. This manuscript should be published soon and we will update the reference accordingly.

Line 169: Is this really deriving the soil carbon "…response to temperature change"? I think the link to temperature change is solely based on interpretation, not model derivation.
Response: This is correct, we have removed the last part of the sentence.

Line 184: Not a paper strictly on soil CO2, but Slessarev et al., 2016 (Nature) might be useful here for linking parameters of the soil carbonate system to the water balance.
Response: We thank the reviewer for this suggestion and will add the reference in the next version of the paper.

Line 194: "…by linear regression of the summer cave monitoring data". I assume these are the large-diamond points in Figure 2. But looking at figure 2 I assume that the regression data are spring, summer, and fall (since monitoring is said to be monthly and there is no indication that spring/fall data are removed). Which data are actually used in the regression?

Response: We apologise for the confusion. The monitoring data used are from May-November, reflecting "summer" conditions when surface bioproductivity is high and cave $pCO_2$ values are elevated. We will clarify this in the figure caption and text.

Line 196: While I suspect the offset of the forest data may actually be due to mixing with modern pCO2 (not pre-industrial levels), if the authors wish to keep this turbulence/advection effect argument I think it is important that a reasonable hypothesis for the signature of the third, unaccounted for air mass is added. Based on atmos circulation and likely boundary layer $\delta^{13}C$ in upstream ecosystems, is this mixing trend reasonable?
Response: As discussed above, we will now add more details, including a post- and pre-industrial mixing line to the figure and update the text to clarify our choice of end members.

Line 201: "… but they provide the best available constraints on the end-member". Wouldn't directly measuring soil CO2 provide a better constraint? (Although, as stated above, I disagree with using a modern calibration to get a Holocene end member)
Response: Directly measuring soil $pCO_2$ at the site would provide the most direct constraint on present-day composition of the respired end member. However, since our study calibrates the mixing line for a pre-industrial scenario, this would not be useful in our case.

Line 217: Why was each simulation for each timeslice repeated twice? Were they varied from one simulation to the other? (Table 1 only gives single values for each timeslice)
Response: Apologies, this is probably phrased confusingly. What is meant is that for each combination of all parameters, two simulations were performed, once using EH values for atmospheric $CO_2$ (concentration and $\delta^{13}C$) and temperature, and once using LG values. These were derived from the literature. However, our new approach now calculates the mixing lines differently depending on the time period, so this step is not necessary anymore.

Line 221-226: I'm a bit confused. Is there one set of binary filtering for the three mixing line simulations, and a different filtering approach (just selecting the best 5%) for the sensitivity tests?
Response: Yes, this is correct. It was necessary to use different approaches since the sensitivity test would not lead to matching solutions within the constraints of the proxy uncertainties, as there is not enough variation in the $\delta^{13}C$. On the other hand, for the mixing line experiments, using the constraint of the best 5% does not appropriately constrain the solutions and leads to excessive spread of the data.

Figure 4: Are measurement uncertainties considered in these regressions?
Response: No measurement uncertainties were considered here as these regressions are mainly meant to illustrate the concept. We will add information on the uncertainties in the next version of the manuscript.

Line 293-294: I don't think that encouraging model results is confirmation that "the estimate of the soil respired end member composition is accurate". More sensitivity tests are needed to demonstrate that other soil respiration end member compositions lead to problematic results (particularly when the end members are allowed to vary with time, as discussed above).

Response: We have now performed additional sensitivity tests and added changing atmospheric and respired end member compositions. These results do not significantly change the main findings of our study.

Line 309-311: This is another instance where I'm tripped up by terminology. I think "initial soil gas" is the same as "soil respired CO2" and not the same as just "soil gas" or "soil CO2"?
Response: As discussed above, this reflects the parameter selection used for the model, where we calculated our mixing lines that reflect a high $pCO_2$ end member (but not reflecting respiration only). Therefore, we use the term "soil gas". We will clarify this in the text.

Line 310-311: I would like to know more about this. Why does the sensitivity test require such an enriched $\delta^{13}C$ end member? How is the use of this end member justified over the use of the "calibrated" one? Is it a problem that the -22 per mille value does not yield many positive results?
Response: Our tests have shown that to obtain such high stalagmite $\delta^{13}C$ values as recorded during the last glacial, the soil gas $\delta^{13}C$ that the solution equilibrates with also has to be quite high. With the mixing lines, this results in the selection of an initial soil gas composition that has a larger atmospheric component (i.e., higher $\delta^{13}C$ and lower $pCO_2$). With the sensitivity test, this is not possible, as we keep $\delta^{13}C$ fixed. To avoid having too many simulations not matching the data (as would be the case when using a very negative end member), we opted to use a more enriched end member. This is not a problem, but rather illustrates how the speleothem $\delta^{13}C$ trend over the deglaciation requires a change in the initial soil gas $\delta^{13}C$.

Line 328: What is meant by "depth" here? I don't think the Pataki paper actually measures anything over soil depth.
Response: We have removed this expression.

Line 347: This is probably just my own problem, but I'm confused with terminology again. I thought initial soil gas might be the initial CO2 from soil respiration, but this sentence implies initial soil gas and soil respired CO2 are two distinct things.
Response: See comment for lines 309-311.

Line 433: "…best explained by.." my version of the document just says "c."
Response: Apologies for this mistake, it is now corrected and the sentence reads: "… the temperature sensitivity of $\delta^{13}C_{spel}$ over the last deglaciation in Western Europe is best explained by increasing soil respiration."

**References cited:**
Baldini, L.M., McDermott, F., Baldini, J.U.L., Arias, P., Cueto, M., Fairchild, I.J., Hoffmann, D.L., Mattey, D.P., Müller, W., Constantin, D., Ontañón, R., Garciá-Moncó, C., Richards, D.A., 2015. Regional temperature, atmospheric circulation, and sea-ice variability within the Younger Dryas Event constrained using a speleothem from northern Iberia. Earth Planet. Sci. Lett. 419, 101–110. doi:10.1016/j.epsl.2015.03.015
Borsato, A., Frisia, S., Miorandi, R., 2015. Carbon dioxide concentration in temperate climate caves and parent soils over an altitudinal gradient and its influence on speleothem growth and fabrics. Earth Surf. Process. Landforms 40, 1158–1170. doi:10.1002/esp.3706

Denniston, R.F., Houts, A.N., Asmerom, Y., Wanamaker Jr., A.D., Haws, J.A., Polyak, V.J., Thatcher, D.L., Altan-Ochir, S., Borowske, A.C., Breitenbach, S.F.M., Ummenhofer, C.C., Regala, F.T., Benedetti, M.M., Bicho, N.F., 2018. A stalagmite test of North Atlantic SST and Iberian hydroclimate linkages over the last two glacial cycles. Clim. Past 14, 1893–1913.

Genty, D., Blamart, D., Ghaleb, B., Plagnes, V., Causse, C., Bakalowicz, M., Zouari, K., Chkir, N., Hellstrom, J., Wainer, K., Bourges, F., 2006. Timing and dynamics of the last deglaciation from European and North African $\delta^{13}C$ stalagmite profiles - Comparison with Chinese and South Hemisphere stalagmites. Quat. Sci. Rev. 25, 2118–2142. doi:10.1016/j.quascirev.2006.01.030

Genty, D., Blamart, D., Ouahdi, R., Gilmour, M., Baker, A., Jouzel, J., Van-Exter, S., 2003. Precise dating of Dansgaard-Oeschger climate oscillations in western Europe from stalagmite data. Nature 421, 833–837. doi:10.1038/nature01391

Moreno, A., Stoll, H., Jiménez-Sánchez, M., Cacho, I., Valero-Garcés, B., Ito, E., Edwards, R.L., 2010. A speleothem record of glacial (25-11.6 kyr BP) rapid climatic changes from northern Iberian Peninsula. Glob. Planet. Change 71, 218–231. doi:10.1016/j.gloplacha.2009.10.002

Pataki, D.E., Ehleringer, J.R., Flanagan, L.B., Yakir, D., Bowling, D.R., Still, C.J., Buchmann, N., Kaplan, J.O., Berry, J.A., 2003. The application and interpretation of Keeling plots in terrestrial carbon cycle research. Global Biogeochem. Cycles 17. doi:10.1029/2001GB001850

Rossi, C., Bajo, P., Lozano, R.P., Hellstrom, J., 2018. Younger Dryas to Early Holocene paleoclimate in Cantabria (N Spain): Constraints from speleothem Mg, annual fluorescence banding and stable isotope records. Quat. Sci. Rev. 192, 71–85. doi:10.1016/j.quascirev.2018.05.025

Scheff, J., Seager, R., Liu, H., Coats, S., 2017. Are Glacials Dry? Consequences for Paleoclimatology and for Greenhouse Warming. J. Clim. 30, 6593–6609. doi:10.1175/JCLI-D-16-0854.1

Schmitt, J., Schneider, R., Elsig, J., Leuenberger, D., Lourantou, A., Chappellaz, J., Köhler, P., Joos, F., Stocker, T.F., Leuenberger, M., Fischer, H., 2012. Carbon isotope constraints on the deglacial $CO_2$ rise from ice cores. Science (80-. ). 336, 711–715.

Stoll, H.M., Müller, W., Prieto, M., 2012. I-STAL, a model for interpretation of Mg/Ca, Sr/Ca and Ba/Ca variations in speleothems and its forward and inverse application on seasonal to millennial scales. Geochemistry, Geophys. Geosystems 13, 1–27. doi:10.1029/2012GC004183

Verheyden, S., Keppens, E., Quinif, Y., Cheng, H., Edwards, L.R., 2014. Late-glacial and Holocene climate reconstruction as inferred from a stalagmite - Grotte du Pere Noel, Han-sur-Lesse, Belgium. Geol. Belgica 17, 83–89.

Waring, C.L., Hankin, S.I., Griffith, D.W.T., Kertesz, M.A., Kobylski, V., Wilson, N.L., Coleman, N. V, Kettlewell, G., Zlot, R., Bosse, M., Bell, G., 2017. Seasonal total methane depletion in limestone caves. Sci. Rep. 7. doi:10.1038/s41598-017-07769-6

---

## Author Comment (AC3)

**Response to anonymous referee #2**

**General comments**
This manuscript combines multi-proxy analyses ($\delta^{13}$C, $\delta44$Ca, paired U-Th and 14C ages) and geochemical modelling of vegetation-soil respiration within the soil-karst-cave system in the northern Iberian Peninsula over the last glacial-to-interglacial transition (ca. from 26 ka to 4 ka).

Authors have irrefutable knowledge on the subject and are familiar with the region, data and tools applied. They have performed comprehensive analytical work on three speleothems from two caves [Candela, El Pindal Cave (Moreno et al., 2010; Rudzka et al., 2011; Stoll et al., 2013; this study); Laura, El Pindal Cave (this study); Galia, La Vallina Cave (Stoll et al., 2013; this study)], including three pieces of the overlying bedrock.

The focus is on the $\delta^{13}$Cspeleo signal (Fohlmeister et al., 2020) to decipher whether it can be a possible paleo-soil respiration proxy (pCO2). CaveCalc (Owen et al., 2018) and ISTAL (Stoll et al., 2012) are used to account for effects of prior calcite precipitation (PCP), mean soil carbon age - dead carbon fraction (DCF), karst hydrology, bedrock dissolution, seepage zone and drip interval length changes. The referential temperature pattern is taken from Iberian Margin marine sediments (Darfeuil et al., 2016), the radiometric chrono-stratigraphy of which is sufficiently robust, and multiproxy studies have been performed on its strata.

Results point to increasing soil pCO2 over the last deglaciation (from late glacial ca. 530-1030 ppmv to ca. early interglacial 1155-5780 ppmv) heavily dependent on temperature (Q10 ~ 2.7-7; i.e., a factor by which soil respiration increases with a 10°C rise in temperature). This is in line with previously documented changes in vegetation cover and substrate from open glacial grassland, steppe taxa and low arboreal percentages to interglacial high arboreal pollen (data compilations in Fletcher et al., 2010 and Moreno et al., 2014; simulations in Scheff et al., 2017). Authors present and discuss other possible processes which are not found to exert a huge impact on the δ13Cspeleo signal. Their results, interpretations and conclusions are justified by data and are consistent with previous monitoring data that showed seasonal variations in cave pCO2 driven by external temperature variations (Moreno et al., 2010; Stoll et al., 2012).

The science of the manuscript is excellent (significance and quality) and the overall presentation well structured. The organisation and length of the manuscript are good: 1 Table and 6 Figures in main text and 4 appropriate supplementary Figures (see below for specific comments). The title clearly reflects the contents of the paper and the abstract provides a concise and complete summary. The subject addresses relevant scientific questions within the scope of CP. My opinion is that it merits publication with minor changes once few clarifications are added. Please see below for constructive suggestions.

Response: We thank the reviewer for these supportive comments and provide more detailed responses to their suggestions for improvement below.

**Specific comments and technical corrections**

**Tables**
Suppl. Table 1, Suppl. Table 2
I was unable to find the 2 supplementary Tables mentioned in the text.

Response: Apologies for this, we will provide the tables with the next version of the manuscript.

**Figures**
**Current Figure 1 (Speleothem δ13C records covering the last deglaciation in temperate Western Europe)**
- Remove lines when 'hiatus' or 'no data' in panels A, C
Response: We believe the reviewer refers to the record from Villars cave (blue), which is very low resolution and therefore looks odd, although there is no hiatus. We will add a sentence clarifying this to the figure caption to avoid further confusion.

- Change "El Pindal (study site)" to "El Pindal & La Vallina (this study)" in legend of panel A
- Complete figure caption. Something like: "El Pindal Cave – stalagmite Candela (Moreno et al., 2010, this study), – stalagmite Laura (this study) and La Vallina Cave – stalagmite Galia (Stoll et al., 2013; this study)."
Response: The records shown here are only previously published studies, therefore we are now showing the new data from Laura and Galia. We will update the caption to clarify this.

- δ13C Villars (Genty et al., 2006; Wainer et al., 2011) does not appear to be consistent with El Pindal ca. 19 ka. Issue with resolution of the former? Any comment?
Response: We are not sure why the Villars record deviates from El Pindal and Chauvet before 19 ka. The issue is specifically with the record Vil-car1 (Wainer et al., 2011), which does indeed seem to show a different behaviour than other records from Villars cave. The authors of the Vil-car1 paper note that the record appears significantly affected by disequilibrium isotopic fractionation, which might explain the discrepancy.

- δ13C Buraca Gloriosa (Denniston et al., 2018) appears opposite to El Pindal (Moreno et al., 2010) ca. 13 ka and around 19 ka. Dating issues? Other reasons?
Response: Again, we are not sure about the reason for these high frequency discrepancies. The Buraca Gloriosa $\delta^{13}C$ record is interpreted in the same way as El Pindal, but it is possible that chronological uncertainty leads to the observed discrepancy.

- δ13C Cova da Arcoia (Railsback et al., 2011, PPP 305) is not included. It seems to have quite different absolute values during the time span Galia grows (ca. 9 ka). Any comment? Could this be relevant to the comparison with temperatures derived from the marine record located further south? Atlantic versus Mediterranean climatic zones? (see for instance Fig. 1 in Denniston et al., 2018)
Response: Our study focuses on caves in settings where soil $pCO_2$ is temperature limited, and therefore we argue that our findings have more broader significance than the specific region of northern Iberia. The reviewer makes a good point that stalagmites from caves located further south on the Iberian Peninsula have different trends in $\delta^{13}C$, as here soil $pCO_2$ will likely be moisture limited, leading to very different phasing over glacial-interglacial cycles than temperature. We will make this clarification in the next version of the manuscript.

- In this regard, what about adding δ13C La Mine (Genty et al., 2006) as a contrasting environment?
This is important to highlight the "regional" extent of the exercise submitted in the present study.

Response: The reviewer is correct that our study focuses on a specific region, but we would like to emphasize that our findings do have broader impact, as they will apply to any system where soil $pCO_2$ is temperature limited. Moreover, we combine the use of $\delta^{13}C$, DCF and $\delta^{44}Ca$ measurements to better constrain initial soil conditions, which is a novel use of the speleothem archive. We will clarify this in the next version of the manuscript, but we think adding another record to figure 1 that is not from the Western European region would add confusion.

**Additional Figure (new Figure 1? Current numbers would change accordingly up to 7 Figures)**

A non-specialist reader would very much appreciate being able to recognise all the variables measured and modelled under discussion. Thus, I earnestly request that authors include an illustrative scheme with the processes and reactions in question. As far as possible, the text must be self-explanatory: labelling the parameters as in the Figures, i.e. δ13C, pCO2, specifying sources and including the notion of dead carbon (modern-to-fossil reservoir effect) so the reader can follow the reasoning step by step: (i) atmospheric CO2 and rainwater (highlight seasonal effects in temperature and rainfall density/amount); (ii) biogenic CO2 from vegetation-plant litter-microbial activity-soil respiration (emphasize role of moisture availability, vegetation type and cover in soil gas pCO2); (iii) infiltration through soil water to karst and water flow paths, bedrock dissolution, drip water, CO2 degassing-calcite precipitation to form the speleothem (link to cave air pCO2, cave ventilation dynamics, etc). Perhaps two panels are needed: one for a 'summer' scenario (assimilated to interglacial situation?) versus a 'winter' one (for the glacial conditions?). Ideally, this must lead the reader through the diverse situations deduced from the results, without digging too much in dispersed literature, which is somewhat scarce for δ13C specifically (indeed this is a strong point of the present manuscript value). Consider adding the seasonality of the caves in question with the series of instrumental measures available for the area (see below for additional comments on that).

Response: We appreciate this comment from the reviewer, and we agree that it is difficult to keep track of all variables under discussion when modelling carbon isotopes. Rather than a new schematic figure here, we propose to refer to similar figures that have previously been published that we can refer to in the text, e.g. in Rudzka et al., (2011), or in Mattey et al., (2016).  Instead, we suggest that the main points evaluated here would be clarified by the inclusion of a set of simple figures illustrating the isolated influence on speleothem $\delta^{13}C$ of changes in a) the degree of openness of dissolution, b) the effect of soil $pCO_2$ given a constant dead carbon fraction, and c) the effect of prior calcite precipitation (Figure 1). This figure will serve as a useful reference point to comment how independent proxy record of DCP from $^{14}C$ allow constraining the open system effect, and how independent constraints on PCP from $\delta^{44}Ca$ allow constraints on PCP effect, making it possible to narrow the range of possible soil $pCO_2$ effects on speleothem $\delta^{13}C$.

[Figure]

*Figure 1: Processes dominantly affecting stalagmite $\delta^{13}C$. a) Carbonate bedrock dissolution under open-closed system conditions (shown by percent of dead carbon added to solution), b) soil pCO$_2$ that the solution equilibrates with, c) prior calcite precipitation, i.e., how much carbonate is precipitated before reaching the stalagmite. The dissolution process can be constrained using DCF, while PCP is constrained using $\delta^{44}Ca$. Soil pCO$_2$ affects all three proxies, but can be constrained further using the coupled relationship with $\delta^{13}C$ from the mixing lines.*

**Figure 3**
- Complete figure caption to highlight the paired U-Th and 14C ages shown at the bottom of the figure. Refer to the Suppl. Table 2 (if it exists?)
Response: Thank you for this suggestion, we will add a note to the figure caption and refer to Suppl. Table 2.

**Figure 6**
This figure seems too compacted. Try to uniform the criteria for all Figures, so the period of interest (from 26 ka to 4 ka?) and the relevant events of the study are clear.
Response: We will update this figure accordingly.

**Main text**
**References are made to the text by giving [line numbers: "text quotes"].**

[Line 15 "underwent dramatic climatic and environmental change"]
Please remove "dramatic". If qualifying the change is needed, any alternatives? "profound" is used for the Introduction, what about "significant" here?
Response: Done

[Lines 17-19 "global carbon cycle" … "on local soil respiration"]
My recommendation is that neither the word "global" nor the point to "local" fit in here or at least may add confusion. The present work may have "regional" application (and unvaluable as such!)  for similar temperate environments of Western Europe when results are properly reproduced in subsequent studies.
Response: Thank you for this suggestion. We have removed the term "local" from the sentence, but retained "global carbon cycle" as this refers to the significance of understanding soil respiration at a global scale.

[Lines 21, 73, 88, 92, 325, 336, 337, 349 … "Northern Spain", "NW Spain", "northern Spain"…]
Check for consistency and consider changing to geological terms such as "NW Iberian Peninsula".

Response: Thank you for the suggestion. We have changed the term to "NW Iberian Peninsula" throughout the text.

[Lines 34-35 "Between 22 and 10 ka BP (ka: thousands of years, BP: "before present", with the present referring to 1950 CE),"
[Lines 103-104 "Minimum average temperatures are reconstructed for Heinrich event 1 (H1; 18-15 ka BP) and are ~8°C cooler than those of the Holocene Thermal Maximum (~8 ka BP; Darfeuil et al., 2016)."]
[Lines 246, 252, 256, 260, 298, 301, 333, 427 … "LGM (26.8 ka BP)" "the LGM (24 ka BP)" "(LGM, H1, and YD)" "during the LGM and YD" "~530-1030 ppmv during the LGM, and ~1155-5780 ppmv during the EH"]
[Line 316 "for the Early Holocene (EH, post 10 ka BP) and the Late Glacial (LG, pre 10 ka BP and including deglacial)"]
These excerpts use terms and chronostratigraphic units that must be clarified.
For instance, "last glacial maximum" (LGM) is used, though I am afraid I do not find the complete acronym meaning anywhere in the manuscript. In any case, both characterisation and timing of the LGM are complex enough for including the term here (see different approaches and stratigraphy ranging from ca. 33 ka to 26.5 or 23 ka to 19 ka, depending on literature e.g., Peltier & Fairbanks, 2006, QUAT. SCI. REV. 25; Clark et al., 2009, SCIENCE 325; Batchelor et al., 2019, NATURE COMM. 10; Gowan et al., 2021, NATURE COMM. 12; and references therein). Decoupling between temperatures and ice volume is specifically pronounced during deglaciations. Temperature estimations at the Iberian Margin suggest that the LGM was not a real stadial but a kind of weak interstadial. Although undoubtedly cold, it was not the coldest interval. The coldest intervals are observed during Heinrich events. Following the reference used in the manuscript (Lambeck et al., 2014), the main phase of deglaciation occurred from ca. 16.5 ka to 8.2 ka. My advice would be to delete any reference to "the LGM" and stick to two phases Late Glacial (LG) and Early Holocene (EH). Similarly, avoid the reference to a "Holocene Thermal Maximum", which is an even more diffuse designation. The "Holocene temperature conundrum" debate will likely remain highly contentious over many years to come (Liu et al., 2014, PNAS 111; Bader et al., 2020, NATURE COMM. 11; Martin et al., 2020, QUAT. SCI. REV. 228; and references therein).
Additionally, the base of the Holocene must be placed ca. 11.7 ka, not 10 ka (Walker et al., 2009, J. QUAT. SCI. 24) and the EH spans from 11.7 ka to 8.2 ka (Greenlandian; Walker et al., 2019, J. QUAT. SCI. 34), though technically speaking the present study shows results up to 4 ka in Fig. 3, i.e. the Mid-Holocene (Northgrippian; Walker et al., 2019, J. QUAT. SCI. 34). This does not alter the results of the manuscript but respects the formal definition and dating established, in line with the useful INTIMATE event stratigraphy of Greenland interstadials and stadials (GI and GS, respectively; Lowe et al., 2008, QUAT. SCI. REV. 27; Rasmussen et al., 2014, QUAT. SCI. REV. 106; Mojtabavi et al., 2020, CP 16, 2359). For the LG events, please consider this nomenclature (i.e., use GS-1, not YD; and GS-2.1a, not H1), which implies showing a Greenland d18O profile in the Figures where these intervals are discussed. These are aspects of relevance to the subject because, the manuscript works on and paves the way to well dated speleothem material, with chronologies specifically reviewed within the SISAL database, version 2 (Comas-Bru et al., 2020a,b).

Response: We thank the reviewer for this in-depth comment. We apologise for the inconsistencies in terminology and chronostratigraphic units used in the text, and we agree that it is best to stick with the LG/EH time slices. Given our updated modelling framework with different mixing lines accounting for changes in atmospheric $CO_2$ over the glacial-interglacial transition, we now use three time slices: LG for the period older than 24 ka, DEG for the deglacial transition (15-11.7 ka), and EH for the period younger

than 11.7 ka. We have also changed the nomenclature for the LG events as suggested, and show the Greenland $\delta^{18}O$ record together with the Iberian Margin SST record in figures 1 and 3.

[Lines 96-99 "(AEMET meteorological stations at Santander and Oviedo, period 1973-2010; AEMET, 2020)" "(AEMET meteorological station at Santander, period 1987-2000; AEMET, 2020)"]
[Lines 469-470 "AEMET, 2020. State Meteorological Agency (AEMET) [WWW Document]. URL http://www.aemet.es/en/portada (accessed 470 10.8.20)"]
Not sure I understand the data source used here. Are the time intervals 1973-2010, 1987-2000 chosen for a particular reason? Is there a gap between 2010 and 2020? Can the series be shown for instance in the new Figure? Something that illustrates the seasonality in the region and explains more clearly the assumptions for the parameters involved in the present study (cave-monitored CO2, $\delta^{13}C$, etc).
Response: This refers to the governmental meteorological agency data. We will add more clarifications with respect to the effects of seasonality on soil and cave parameters to the text and add the new figure (Figure 1).

[Lines 98, 101-102, 106-107, 197-198, 407-409 "winter months (December-February)", "summer months (June-September)" "estimate of the deglacial temperature change in caves on the coastal plain, as the region's modern seasonal cycle displays similar amplitude to sea surface temperatures (Stoll et al., 2015)." "caves are well ventilated in the cold season with close to atmospheric pCO2 values, but feature elevated CO2 concentrations during the warm summer season (Stoll et al., 2012)." · "Cave monitoring data from winter months (December-March) were excluded from the regression analysis"; "Two model scenarios mimick full glacial and Holocene conditions, including changes in temperature, cave pCO2, and soil pCO2 for "winter" (i.e., atmospheric) and "summer" (i.e., elevated) cave pCO2 (Suppl. Fig. 4).]]
Response: Thank you for spotting this. We will refer to the months more clearly in the next version of the manuscript.

Please correct "mimick" to 'mimic'; or better still, change the word to "simulate"?
Response: Done.

Seasonal changes, both in CO2 and temperature, appear crucial for interpretation of the results. Please clarify as much as possible throughout the manuscript. This would improve if illustrated with the new Figure. The reader would appreciate a clearly understandable and comprehensive discussion on that. For calibration purposes, I wonder if databases considering non global atmospheric CO2 values but continuous seasonal CO2 measurements from the ground-based network ICOS may be of some assistance here (Integrated Carbon Observation System, ICOS; Ramonet et al., 2020, Phil. Trans. R. Soc. B 375). Any comments?
Response: Thank you for this comment and the suggestion to use regional $CO_2$ measurements. As suggested by reviewer 1 we do now include the measured modern (rather than Suess-corrected) $\delta^{13}C$ data and a modern atmospheric end member. However, the seasonal variations in atmospheric $CO_2$ are small relative to the range along the mixing line, and we find that using seasonally resolved modern atmospheric composition does not have any appreciable effect on the estimation of the modern respired end member. This use of a global $pCO_2$ is also consistent with the approach we must take for the speleothem modeling, as we use ice core estimates of global $pCO_2$ from glacial to early Holocene time periods.

[Lines 103-104, 343 "Heinrich event 1 (H1; 18-15 ka BP) and are ~8°C cooler than those of the Holocene Thermal Maximum (~8 ka BP; Darfeuil et al., 2016)." "Assuming a temperature change of roughly 8°C between the LGM and EH (Darfeuil et al., 2016)"]
Please clarify. It seems the 8⁰C value accounts for the increase of temperatures between GS-2.1a (H1; ca. 18-15 ka) and the EH (before 8.2 ka). Other alternatives, i.e., from LGM to values after 8.2 ka seem closer to 6⁰C, though perhaps I am missing something here. I understand the selection criteria of the site used as a reference for temperature (Iberian Margin site MD95-2042; Darfeuil et al., 2016) is based on its chrono-stratigraphy? I'd suggest authors also highlight the fact that multiproxy studies have been performed on its strata. In Darfeuil et al., 2016, two complementary paleo-thermometers are discussed, the TEX86 and Uk'37 (annual mean sea surface temperatures, a potential shift towards summer production that may occur for glacial times?). Authors refer to the former only and the profile is shown in Fig 3. Any comment here considering seasonality? Please include considerations of the analytical and calibration errors of the estimates. What about alternative documentation provided by pollen transfer functions? Perhaps it would be preferable to have a sediment core further north, closer to the caves, though to my knowledge this is not available.
Response: Thank you for this comment. We will address this comment in the next version of the manuscript and clarify the underlying issues with these reconstructions (analytical and calibration errors, seasonality). It would indeed be very nice to have a sediment core closer to the caves, but as the reviewer points out, this is not available.

[Line 129 "Stoll et al., in review"].
If the paper is not publicly available at the time the present manuscript is published, I would suggest that the authors remove the reference in review and point to a different reference already peer-reviewed or add the information in this study.
Response: Thank you. This manuscript should be published soon and we will update the reference accordingly.

[Lines 142.145 "Reimer, 2013"; "Reimer et al., 2013"]
It may be advisable to work on the updated calibration curves, i.e. IntCal20 and Marine20; Reimer et al., 2020, Radiocarbon, 62; Heaton et al., 2020a,b, Radiocarbon, 62. For Marine20, marine reservoir ages are modelled as time-varying, though for IntCal20, speleothem dead carbon fractions are approximately constant over time but with an unknown level. Any comment here?
Response: We agree that in general the use of up to date calibration curves is advisable. However, since changes in the calibration curve over the studied interval are minor (and uncertainties related to chronology in the speleothems will be the dominant source of uncertainty in DCF) we refrain from updating the records.
The calculation of speleothem dead carbon fraction in our study is based on paired U-Th and $^{14}$C measurements, which makes them especially robust as chronological uncertainty from age interpolation procedures is avoided. We are not sure what the reviewer refers to with respect to the dead carbon fractions in IntCal20: it is true that the DCF for the calibration curve intervals beyond atmospheric values is based largely on extrapolation of the well-constrained DCF at Hulu Cave, however, this does not directly affect our reconstruction (except for increased uncertainty in the calibration curve, which then translates to the DCF values).

[Line 360 "vegetation cove"]
Change to "vegetation cover".
Response: Thank you for spotting this mistake, now corrected.

[Line 433 "δ13Cspel over the last deglaciation in Western Europe is best explained by c"]

Please complete the sentence.

Response: Apologies for this mistake, it is now corrected and the sentence reads: "… the temperature sensitivity of $\delta^{13}C_{spel}$ over the last deglaciation in Western Europe is best explained by increasing soil respiration."

**References cited:**

Mattey, D.P., Atkinson, T.C., Barker, J.A., Fisher, R., Latin, J.-P., Durell, R., Ainsworth, M., 2016. Carbon dioxide, ground air and carbon cycling in Gibraltar karst. Geochim. Cosmochim. Acta 184, 88–113. doi:10.1016/j.gca.2016.01.041

Rudzka, D., McDermott, F., Baldini, L.M., Fleitmann, D., Moreno, A., Stoll, H., 2011. The coupled $\delta^{13}$C-radiocarbon systematics of three Late Glacial/early Holocene speleothems; insights into soil and cave processes at climatic transitions. Geochim. Cosmochim. Acta 75, 4321–4339. doi:10.1016/j.gca.2011.05.022

Wainer, K., Genty, D., Blamart, D., Daëron, M., Bar-Matthews, M., Vonhof, H., Dublyansky, Y., Pons-Branchu, E., Thomas, L., van Calsteren, P., Quinif, Y., Caillon, N., 2011. Speleothem record of the last 180 ka in Villars cave (SW France): Investigation of a large δ18O shift between MIS6 and MIS5. Quat. Sci. Rev. 30, 130–146. doi:10.1016/j.quascirev.2010.07.004

---

## Author Response (AR1)

**Response to anonymous referee #1**

This paper presents a useful modeling approach for reconstructing soil respiration using stalagmite carbon isotopes and proxy constraints on prior calcite precipitation and bedrock dissolution effects. Carbon isotopes have been a particularly messy avenue in speleothem science due to the complex interplay of these effects (and others) and the paper represents an exciting effort toward rigorously disentangling this mess. The model is an important step toward understanding how soil respiration changes with climate and, given the breadth of data available in the SISAL database, could be quickly applied on a large scale (assuming it can be appropriately constrained). But the modeling approach carries some critical (and likely invalid) assumptions that need to be addressed. This paper can be of sufficient interest for Climate of the Past and I think my points can be addressed with major revisions. I commend the authors for their coupled proxy-model approach and hope my feedback is useful as they refine their work.

My expertise most closely aligns with the modeling work, so I focus my feedback on this part of the manuscript. I can't speak much to the analytical methods. Below, I've divided my feedback into points about the main modeling approach (outlining my own confusion) and line-by-line items. My biggest concerns are that the modern calibration of the soil respired end member seems invalid (or at least should use modern CO2 levels), and that it is not a safe assumption that the mixing end members are time-invariant.

Response: We thank the reviewer for their thorough and fair assessment of our manuscript. We address the comments in detail below. All changes to the manuscript are highlighted in yellow in the new version.

**Main modeling points**

My understanding of the modeling approach involves three steps: (1) use modern data to derive a relationship between atmospheric CO2 and soil-respired end members; (2) assuming this mixing relationship holds through time prescribe the full range of soil CO2 and  $\delta^{13}$ C possibilities to solve for proxy data with CaveCalc; (3) using the model output, select and analyze the combination of input parameters that yield results closely in line with the measured data. Below, I dive into my concerns on steps 1 and 2 in more detail and include one note on step 3.

**Step 1: calibrating a soil respiration end member**

This calibration exercise (outlined in Figure 2) carries four big assumptions that I think need to be addressed. The biggest has to do with using modern data to calibrate a pre-industrial mixing curve (see point 2).

 First, cave-monitored CO2 and δ13C are used to calibrate a soil CO2 mixing line. This assumes that the mixing of atmosphere and respired CO2 in the cave falls the exact functional form of mixing in the soil (put otherwise, it assumes that the bedrock contribution to cave carbon is the same as soil carbon and that there are no other carbon fluxes distinguishing cave from soil). The authors concede this does not hold true in the winter, but could more be said about this assumption in the months of April-November (when (I think) the monitoring data are used)? Could changes in hydrology, or cave vs soil temperatures, or other things violate this assumption? Is soil CO2 assumed to reflect the soil-column integrated conditions? Is it a problem that this assumption breaks down seasonally if calcite deposition is seasonally biased? I think, at the least, it must be written that this assumption is a bit vague to me). Response: We agree that the assumptions made when calibrating the soil gas end member would benefit from clarification in the text. Here we summarize our reasoning on using cave air measurements to constrain soil gas and estimate the respired end member.

In the revised manuscript we now show the actual monitored data with modern atmospheric composition and post industrial mixing line, which we use to derive the respired end member isotopic composition. Using this end member, we then calculate pre-industrial mixing lines taking into account the atmospheric composition and correcting for the Suess effect to model the soil CO2.

We have added the relevant clarifications to the manuscript (methods section and discussion) and updated figure 2 (Figure 1 in this document). Since we have now used the original monitoring dataset (not corrected for the Suess effect), the resulting respired end member isotopic composition is slightly different, which impacts the modelling (see point 2). We have updated the tables, figures and results section accordingly.

**Estimation of soil respired end member from cave CO**2 **measurements:** In the absence of a full soil monitoring campaign, samples collected from the cave in summer months represent a reasonable approach to estimate the isotopic value of the respired end member contributing to soils/epikarst (Figure 1). This is because the cave, like the soil, is defined by a two-end member mixing system, which is driven by the physical ventilation of the cave. The main fluxes of carbon in a system like El Pindal and La Vallina caves are from soil gas (mainly seeping through the host rock and into the cave) and atmospheric air (through ventilation). Like many mid-and high latitude cave systems, there is a seasonal reversal in the airflow direction in La Vallina cave (Stoll et al., 2012). In the summer, when cave air is colder than exterior air, cave air flows out the entrance, and is replaced by inflow and diffusion of soil/epikarst air. In this season, the cave has the highest CO2 concentrations and points which fall closer to the soil respired end member on the Keeling plot. In the winter, when cave air is warmer than exterior air, exterior air flows in through the cave entrance, bringing the cave closer to the atmospheric end member.

The data from the monitoring of the cave reflects primarily  $CO_2$  from the soil that is drawn through the karst network into the cave. It is therefore likely reflecting soil column-integrated conditions and the full contribution of respired  $CO_2$  in the soil and epikarst unsaturated zone (below the soil, "ground air").

Any contribution of C flux from bedrock dissolution does not significantly affect our estimation of the respired end member, because the intercept defining the respired end member is most influenced by the summer season cave air  $pCO_2$  data. During the summer season drip flow rates are more than an order of magnitude lower than in the winter and degassing from this drip is suppressed by the high cave air pCO2. In winter, when drip rates are higher, and cave air pCO2 is lower, then degassing may contribute to C in cave air, as seen in other systems (Waring et al., 2017). However, winter data corresponding to ventilated periods are near to the atmospheric composition, and we explicitly define our atmospheric end member from global measurements, not the local cave measurements. Hence, the ventilated season measurements, which in theory may have contribution from degassing of dissolved limestone, would have an insignificant impact on our calculated mixing line. Furthermore we do not find evidence for a different intercept in winter and summer end members, unlike monitoring studies which infer a strong effect of degassing of a carbon source from limestone dissolution (Waring et al., 2017). (We also note that contribution of C to stalagmites from bedrock dissolution is explicitly accounted for by modeling of the dead carbon fraction, DCF, in stalagmite data in CaveCalc.)

We therefore argue that summer cave air can be used to estimate the isotopic composition of the respired end member of soil  $CO_2$ , when the competing fluxes are minimized: ventilation is reduced during the summer months (Stoll et al., 2012) and the higher cave air p $CO_2$  levels reduce the amount of degassing that can occur from dripwaters entering the cave (reducing the contribution from host rock carbon). An estimation of the modern respired end member, defined along a mixing line which includes the modern global atmospheric end member, is -26.9 +/- 0.8 ‰. We have included these considerations in the discussion (lines 358-382).

Figure 1: A - Keeling plot of cave and local forest atmospheric CO2. The respired end member is defined through linear regression of the entire dataset. B – Mixing lines defined for the model simulations of past soil gas  $pCO_2$  and  $\delta^{13}C$ . We define three mixing lines based on the changes in the

atmospheric composition (EH, DEG, LG). All three mixing lines use the same respired end member, with a variability of +/- 3 ‰ to account for changes in respired substrate. Biome-level isotopic values for relevant vegetation compositions are shown on the side (from Pataki et al., 2003).

**Seasonally biased calcite deposition**

The seasonality of the modern cave air has the advantage of helping to define the modern respired end member. From our monitoring, we do not find evidence for a different isotopic value of the respired end member in different seasons. Thus, exploiting the seasonal ventilation of the cave to define the mixing line and respired end member does not preclude using this respired end member to interpret records from speleothems in which deposition is dominant in one season. Monthly monitoring of drip rate, dripwater chemistry, and cave air composition indicates that currently in the cave, some stalagmites grow at similar rates throughout the year, whereas in other sectors of the cave, growth occurs exclusively in winter, driven by cave ventilation. We now discuss this aspect in the discussion of the manuscript (lines 383-388).

2. Second, the model is calibrated to pre-industrial CO2 levels even though the data are taken in modern conditions (when CO2 is very well-constrained and much higher!) I think this is done so the same calibration end members can be applied throughout the Holocene (more on this in the "Step 2" section). The forest data are ignored because they might be influenced by "turbulence and advection effects" (Line 196),ut they would probably fit really nicely on a mixing line that reaches to a modern CO2 end member (with higher CO2)! In fact, all of the data would likely fit better on such a line (given that the monitoring data residuals to mixing line 1 are mostly above the line when CO2 > 1,000 ppm). This makes sense, because the monitored data are mixing with the modern atmosphere, not the pre-industrial atmosphere. (Correcting for the Suess effect only corrects for pre-industrial  $\delta^{13}$ C, it does not account for the difference in CO2 between then and now). I highly encourage the authors to use a calibration to a modern end member. Otherwise a much more rigorous justification for the pre-industrial end member is needed.

Response: As suggested by the reviewer, we agree that it is useful to illustrate the full derivation of the modern, post-industrial respired end member also accounting for the modern atmospheric contribution, as a first step (Figure 1). As described in our response to point 1, including the modern atmospheric end member leads to an estimated respired end member of -26.9 ‰. This end member may be more negative than the preindustrial end member (which characterized the Early and mid Holocene growth periods of the stalagmites in this study), because atmospheric  $\delta^{13}$ C has decreased by 2 ‰ over the last century due to anthropogenic activities (Suess effect). If the currently respired end member is composed of modern respiration and respiration of young (decadal age) soil carbon pools, the pre-industrial respired end member may have been as much as 2 ‰ heavier, that is closer to -25 ‰. If a significant fraction of the respired pool is older, then the preindustrial respired end member may fall between - 27 and -25 ‰. We favour the less negative estimate because a young age of respired carbon is suggested by the rapid post-bomb spike decrease in 14C in actively growing stalagmites in the cave.

Therefore, we define an updated reference mixing line for the Early Holocene as having a respired end member of -25 ‰ and an atmospheric end member of 260 ppmv and -6.3 ‰, consistent with ice core records. We use a pre-industrial atmospheric composition in the model since this more closely reflects the end member at the time of stalagmite growth. We have made relevant changes to the manuscript and modelling (lines 195-207, see also point 1).

3. Third, I don't know how the soil-respired end member is defined as 7800ppm and - 22.9‰. I imagine that it's an extrapolation of mixing line 1, but why not extrapolate some number other than 7800? Is there some assumption that I'm missing?

Response: We have clarified that we define the isotopic value of the respired end member on the basis of Keeling plots, however by definition the Keeling intercept is not associated with a particular soil  $CO_2$  concentration. We clarify that 7800 ppm  $CO_2$  (now updated to 8000 ppmv, see step 2) is the soil concentration which best simulates the observed speleothem growth rates and isotopic ratios and we note it is consistent with soil gas p $CO_2$  above other caves in comparable settings (e.g. Borsato et al., 2015). We have also performed sensitivity tests with higher soil gas p $CO_2$ , which do not lead to solutions matching the speleothem.

4. Fourth, I think this modeling approach assumes that boundary layer CO2 concentration and  $\delta^{13}$ C (the stuff that diffuses into the soil) also falls on the same mixing curve. This should be stated since at least two things relevant to this study might violate this assumption. First, a shift from no canopy in the last glacial to a canopy when forests appear might lead to a "canopy effect" whereby  $\delta^{13}$ C gets lower than expected for a given pCO2 due to recycling. Second, "turbulence and advection effects" (line 196) that appear to matter during the daytime (probably when photosynthesis is happening) can overprint the simple mixing relationship and propagate down to the soil respired end member. These effects might well be small, but I think the assumption should at least be recognized.

Response: This is a good point, and we have added these considerations to the discussion of the model (lines 370-373).

**Step 2: assuming this holds through time**

This analysis assumes that the mixing slope between soil respired  $\delta^{13}C$  and atmospheric  $\delta^{13}C$  is constant through time. This is not a good assumption because a lot of factors that matter on decadal or longer timescales (i.e. factors that are not captured by the short calibration) violate it by changing end member CO2 but not  $\delta^{13}C$  (or vice versa). For example, if CO2 increases (like it did from LG to EH), then the isotopic composition of CO2 must decrease to keep the end member on the curve, but we know very well that this assumption is violated on paleoclimate timescales (e.g. Schmitt et al., 2012; Science; Figure 1). Similarly, if soil respiration decreases (thus decreasing soil-respired CO2) then the  $\delta^{13}C$  of soil respiration must increase to stay on the mixing curve. I'm not sure if there's a defensible mechanism for this, although it might occur by coincidence if water stress increases vegetation  $\delta^{13}C$  (thus soil-respired  $\delta^{13}C$ ) while decreasing soil respiration. Either way, I am not aware of any mechanistic reason why the end members of the mixing relationship should, themselves, vary along a mixing curve. The end-members, just like the average soil CO2 values that reflect their mixing, should vary over time.

Response: The reviewer is correct that we expect variation in the composition of the end members over time, and we have now included this in our model.

Two factors may contribute to variation in the mixing line over time. The first and most certain factor is due to changes in the concentration of atmospheric  $CO_2$ . The isotopic composition of the atmospheric  $CO_2$  remains within a few tenths of a permil of the Holocene value (Schmitt et al., 2012). Therefore, assuming a constant respired end member, the slope of the mixing line is reduced during periods of lower atmospheric

pCO2 (Figure 1B). As a consequence, for a given soil gas pCO2 value, the isotopic composition of soil gas pCO2 is more negative during the glacial than during the interglacial. This effect is very small (<0.4 ‰) for soil gas pCO2 4000 ppm or higher, but for soil gas pCO2 of 2000 ppm corresponds to a 0.9 ‰ difference and at 1000 ppmv corresponds to a -1.8 ‰ difference. Nonetheless, this is still a small effect compared to the range of  $\delta^{13}$ C resulting from changes in the soil gas pCO2. If anything, this change in the mixing line would attenuate the difference in  $\delta^{13}$ C between a low CO2 glacial soil and a high CO2 interglacial soil. We now use the three atmospheric end members and the different mixing lines for the modelling of each time period, distinguishing between early Holocene (EH, after 11.7 ka BP), deglaciation (DEG, 16.5-11.7 ka BP) and late glacial (LG, before 16.7 ka BP, Figure 1B).

The mixing line may also vary if the respired end member changes. However, there are few constraints on potential changes in the end member. Although there is variation in the respired end member both within and among biomes, the mean respired end member for the potential biomes which may have characterized this site over the last 25 ka - temperate broadleaf, temperate conifer, and boreal - feature mean  $\delta^{13}$ C of respired end members which differ by only 1 ‰ (Figure 1B; Pataki et al., 2003). This suggests that we cannot predict a systematic change in the  $\delta^{13}$ C of the respired end member with changes in the biome. Moreover, the fact that speleothems across Western Europe show very similar trends in  $\delta^{13}$ C over the deglaciation also indicates that highly localised factors that may lead to a strong change in respired  $\delta^{13}$ C without a biome change are unlikely. Consequently, we address the potential for variation in the respired end member by completing a sensitivity analysis of mixing lines which encompass 3 ‰ heavier and lighter respired end members.

We have added these clarifications to the methods section of the manuscript (lines 208-221) and updated figure 2 accordingly.

I think the modeling can still be performed if some significant changes are made (these are just suggestions and other options can be valid too!).

1. Consider using the actual paleoclimate constraints on pCO2 and  $\delta^{13}$ C of CO2 to parameterize the atmospheric end member.

Response: We have now updated the atmospheric end member for multiple time windows (EH, LG, DEG), in accordance with Schmitt et al. (2012). These do not change the conclusions of the study because they lie largely within the sensitivity range of the different mixing lines and are therefore considered (Figure 2).